# Structural basis for lysophosphatidylserine recognition by GPR34

Tamaki Izume[1,4], Ryo Kawahara[1,4], Akiharu Uwamizu[2,4], Luying Chen [3,4], Shun Yaginuma[2], Jumpei Omi[2], Hiroki Kawana[2], Fengjue Hou[2], Fumiya K. Sano [1], Tatsuki Tanaka [1], Kazuhiro Kobayashi[1], Hiroyuki H. Okamoto[1], Yoshiaki Kise[1], Tomohiko Ohwada [3] ✉, Junken Aoki [2] ✉, Wataru Shihoya [1] ✉ & Osamu Nureki [1] ✉

GPR34 is a recently identified G-protein coupled receptor, which has an immunomodulatory role and recognizes lysophosphatidylserine (LysoPS) as a putative ligand. Here, we report cryo-electron microscopy structures of human GPR34-$G_i$ complex bound with one of two ligands bound: either the LysoPS analogue S3E-LysoPS, or M1, a derivative of S3E-LysoPS in which oleic acid is substituted with a metabolically stable aromatic fatty acid surrogate. The ligand-binding pocket is laterally open toward the membrane, allowing lateral entry of lipidic agonists into the cavity. The amine and carboxylate groups of the serine moiety are recognized by the charged residue cluster. The acyl chain of S3E-LysoPS is bent and fits into the L-shaped hydrophobic pocket in TM4-5 gap, and the aromatic fatty acid surrogate of M1 fits more appropriately. Molecular dynamics simulations further account for the LysoPS-regioselectivity of GPR34. Thus, using a series of structural and physiological experiments, we provide evidence that chemically unstable 2-acyl LysoPS is the physiological ligand for GPR34. Overall, we anticipate the present structures will pave the way for development of novel anticancer drugs that specifically target GPR34.

GPR34 is a G-protein coupled receptor (GPCR) that is evolutionarily conserved in vertebrates[1–3] and shows a high degree of homology to P2Y family members. In various species, GPR34 is expressed in a wide range of tissues and cells, including immune cells, such as microglia[4,5], macrophages[6], type 3 innate lymphoid cells[7], platelets, and dendritic cells[8]. Previous studies have indicated that GPR34 is involved in numerous processes, which include the repair of damaged tissues by type 3 innate lymphoid cells[7], activation of microglial phagocytosis[9], neuropathy pain onset[10], dendritic cell survival[8], and suppression of infection[11]. Despite these myriad functions, the essential roles of

GPR34 remain to be elucidated, primarily due to a lack of consensus regarding the identity of endogenous GPR34 ligands. Two previous studies by Kitamura et al.[1]. and Sugo et al.[12]. identified lysophosphatidylserine (LysoPS) as the GPR34 ligand. This finding prompted Makide et al.[13]. to propose renaming GPR34 as $LPS_1$ or LPSR1, similarly as lysophosphatidic acid receptors ($LPA_{1–6}$). Other such LysoPS receptors, including P2Y10 ($LPS_2$) and GPR174 ($LPS_3$), have also been identified[14]. However, the question of whether LysoPS is truly the physiological ligand for GPR34 remains controversial. In particular, Liebscher and colleagues were unable to replicate the finding by Sugo

[1]Department of Biological Sciences, Graduate School of Science, The University of Tokyo, Bunkyo-ku, Tokyo 113-0033, Japan. [2]Department of Health Chemistry, Graduate School of Pharmaceutical Sciences, The University of Tokyo, 7-3-1 Hongo, Bunkyo-ku, Tokyo 113-0033, Japan. [3]Department of Organic and Medicinal Chemistry, Graduate School of Pharmaceutical Sciences, The University of Tokyo, 7-3-1 Hongo, Bunkyo-ku, Tokyo 113-0033, Japan. [4]These authors contributed equally: Tamaki Izume, Ryo Kawahara, Akiharu Uwamizu, and Luying Chen. ✉e-mail: ohwada@mol.f.u-tokyo.ac.jp; jaok@mol.f.u-tokyo.ac.jp; wtrshh9@gmail.com; nureki@bs.s.u-tokyo.ac.jp

et al.[12] that LysoPS activates mouse and human GPR34 in cAMP inhibition assays[11]. Interestingly, however, the same group showed that two GPR34s from carp fish nicely respond to LysoPS in the same assay[2]. Thus, further work is needed to confirm the identity of the physiological ligand for GPR34, and one possible approach is via the structural determination of the GPR34–LysoPS complex.

LysoPS consists of L-serine and fatty acid moieties connected to a central glycerol molecule by phosphodiester and ester linkages, respectively. Our previous ligand structure–activity-relationship (SAR) studies using chemically modified LysoPS have demonstrated that both the serine and lipid moieties are required for GPR34 activation[15–17]. Physiologically, LysoPS is generated when phosphatidylserine-specific phospholipase A₁ (PS-PLA₁) hydrolyses PS at the sn-1 position to produce sn-2 LysoPS (Fig. 1a)[18]; sn-1 LysoPS is then easily formed by non-enzymatic migration of the ester[19]. Thus, both sn-1 and sn-2 LysoPS are found in mammals and are biologically active. Notably, GPR34 regioselectively prefers LysoPS with an unsaturated fatty acid at the sn-2 position[1]. Similar regioselectivity was also observed for other lysophospholipid-sensing GPCRs such as LPA₃[20] and LPA₆[21]. Indeed, we previously performed SAR studies with synthetic LysoPS analogues that mimic sn-1 and sn-2 LysoPS and, in doing so, identified GPR34-, P2Y10-, and GPR174-selective agonists. Moreover, although the in vivo existence and biological activities of sn-3 lysophospholipids remain enigmatic, we

synthesized LysoPS analogues with the sn-3 configuration and found that these show high potency and selectivity for GPR34[22]. In one case, by replacing the fatty acid with an aromatic group, we succeeded in developing a potent and metabolically stable GPR34 agonist, named M1[22] (Fig. 1a). Critically, such sn-3 LysoPS derivatives represent valuable tools and may hold potential as therapeutic agonists of GPR34.

Previous reports have suggested an immunomodulatory role for GPR34 signalling[7,9,11]; however, as noted above, it remains elusive whether LysoPS is a genuine in vivo ligand for GPR34, or how LysoPS activates GPR34 at the molecular level, limiting the drug development of the GPR34-targeting strategy. Here, we report two cryo-electron microscopy (cryo-EM) structures of human GPR34-Gᵢ complex bound to an sn-3 LysoPS derivative and the potent M1 agonist. These structures, combined with results from molecular dynamics (MD) simulations, reveal the regioselectivity of LysoPS, the lipid-mimetic binding of stable agonist, and the distinct G-protein coupling mode.

## Results
### Overall structures
GPR34 agonists used in this study are shown in Fig. 1a. Depending on the fatty acid position in the glycerol backbone, LysoPS molecules are classified into one of three types (sn-1, sn-2 or sn-3)[22]. The IUPAC names with the definitions of the compounds are summarized in the figure

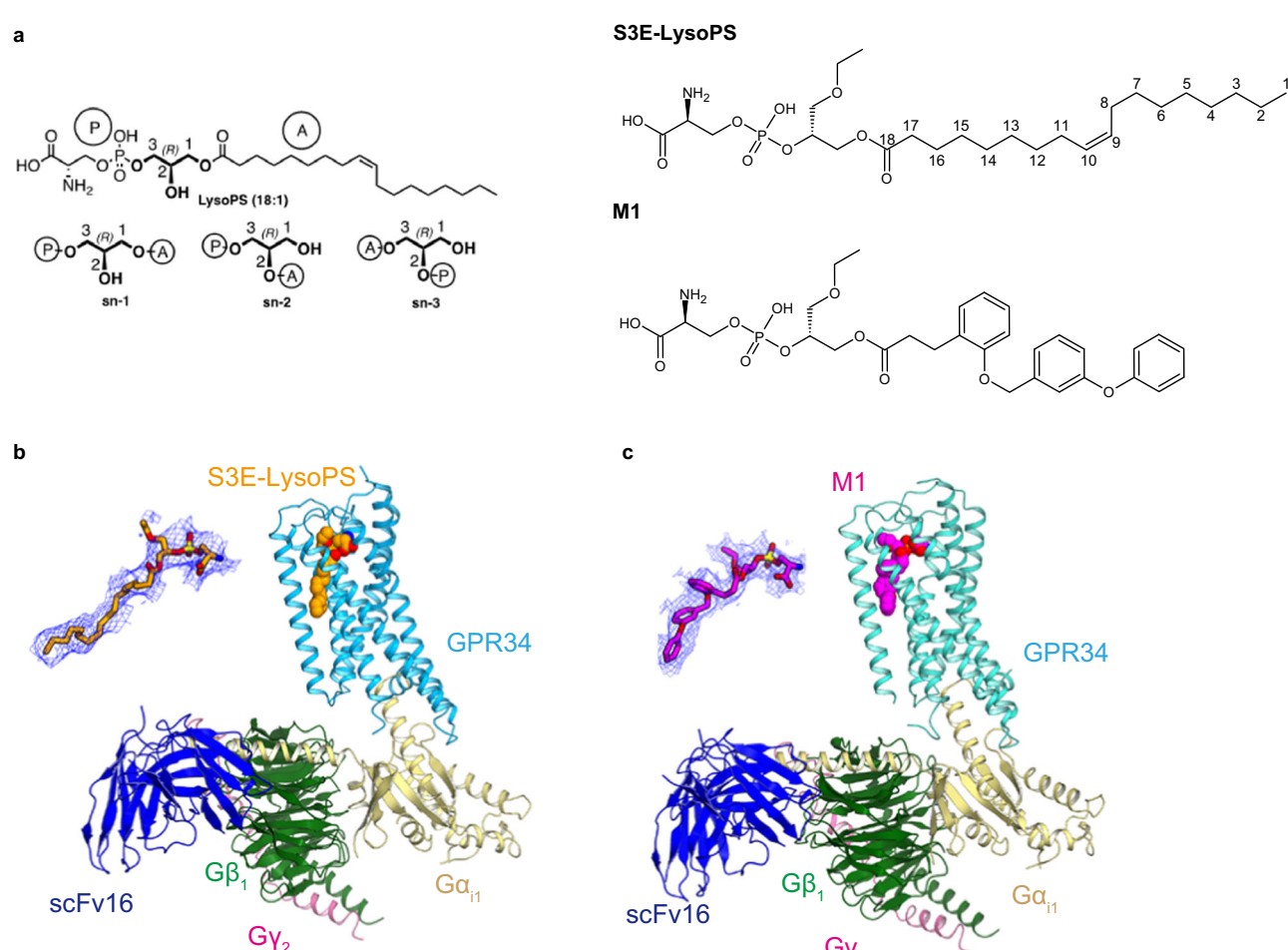

**Fig. 1 | Cryo-EM structures of agonist-bound GPR34-Gᵢ. a** Chemical structures of the GPR34 agonists used in this study. Sn-x (systematic numbering X) is specified for lipids. Sn-1 LysoPS (18:1): O-(hydroxy((R)-2-hydroxy-3-(oleoyloxy)propoxy) phosphoryl)-L-serine (as phosphoserine in sn-1 position and oleoyl chain in sn-3 position) Sn-3 LysoPS (18:1): O-(hydroxy(((R)-1-hydroxy-3-(oleoyloxy)propan-2-yl) oxy)phosphoryl)-L-serine (as phosphoserine in sn-2 position and oleoyl chain in

sn-3 position) M1: O-((((R)−1-ethoxy-3-((3-(2-((3-phenoxybenzyl)oxy)phenyl)propanoyl)oxy)propan−2-yl)oxy)(hydroxy)phosphoryl)-L-serine. **b, c** Overall cryo-EM structures of the (**b**) S3E-LysoPS- and (**c**) M1-bound GPR34-Gᵢ complexes. The agonists are indicated by Corey–Pauling–Koltun (CPK) models, and densities of the agonists are also shown.

legend of Fig. 1a. S3E-LysoPS is an analogue of *sn*-3-type 18:1 LysoPS, with an ethoxy group at the *sn*-1 position. M1 is an analogue of S3E-LysoPS, in which the fragile oleic acid is substituted with a more metabolically stable aromatic fatty acid surrogate (i.e., three tandemly linked phenyl groups, with two ether bonds)[22]. These compounds have been reported to function as potent agonists of GPR34.

Full-length human GPR34 for cryo-EM analysis was expressed in HEK293 cells and purified with S3E-LysoPS or M1. Receptor was then incubated with the $G_i$ heterotrimer ($G\alpha_{i1}$, $G\beta_1$, and $G\gamma_2$) and scFv16, which stabilizes GPCR-$G_i$ complex formation, and the complex was purified by anti-Flag affinity and size exclusion chromatography. We then determined the structures of S3E-LysoPS- and M1-bound GPR34-$G_i$ complexes at nominal global resolutions of 3.3 Å and 2.8 Å, respectively (Fig. 1b, c, Supplementary Fig. 1, and Supplementary Table 1). Local refinement with the mask of the receptor improved local resolution of the extracellular half of the receptor, and the resulting cryo-EM maps allowed modelling of the entire complexes, including agonists (Supplementary Fig. 2).

GPR34 adopts the canonical GPCR topology of a heptahelical transmembrane bundle (7TM), with an extracellular N-terminus, three extracellular loops (ECLs), three intracellular loops (ICLs), and a short amphipathic helix 8 (H8) oriented parallel to the membrane (Fig. 2a). In GPR34, the conserved $P^{5.50}$ (superscripts indicate Ballesteros–Weinstein numbers[23]) is replaced by $I230^{5.50}$, and thus, the transmembrane helix TM5 forms a straight helix. The N-terminus is anchored to TM7 by the disulfide bond $C46^{N-ter}$–$C299^{7.25}$ (Fig. 2b), which is conserved in 15% of class A GPCRs[24,25]. ECL2 (residues 196–213) adopts a U-shape with the TM4–5 side open (Fig. 2b) and is anchored by the disulfide bond $C127^{3.25}$–$C204^{ECL2}$, which is highly conserved in class A GPCRs[26]. ECL2 fills the transmembrane pocket facing toward the extracellular side and provides extensive interactions with TM2–6 (Fig. 2b). Specifically, $F205^{ECL2}$ protrudes into the pocket, and $H206^{ECL2}$ and $K210^{ECL2}$ form salt bridges with $E50^{1.28}$ and $E216^{5.36}$, respectively. The backbone carbonyl groups in ECL2 form hydrogen bonds with residues in TM2, 3, and 6. The tightly packed ECL2 limits the space within the transmembrane pocket and constitutes the ligand-binding site.

## The S3E-LysoPS binding mode

The ligand-binding pocket of GPR34 extends from the centre of ECL2 to the middle of TM4–5, forming an ~25-Å cleft that is laterally open toward the membrane (Fig. 2c). S3E-LysoPS fits into this cleft, oblique to the receptor (Fig. 2c, d). The ligand-binding pocket further consists of both hydrophilic and hydrophobic pockets. The hydrophilic pocket is the canonical GPCR ligand-binding site, composed of TM2, TM3, TM5–7, and ECL2, whereas the hydrophobic pocket consists of TM4 and TM5. The head group and the acyl chain of S3E-LysoPS fit within the hydrophilic and hydrophobic pockets, respectively.

$Y135^{3.33}$, $F205^{ECL2}$, $Y207^{ECL2}$, and $Y289^{6.58}$ create the bottom and sides of the hydrophilic pocket (Fig. 2e), with tilted T-shaped π-π stacking between $Y207^{ECL2}$ and $Y289^{6.58}$. The head group of S3E-LysoPS fits into the pocket, with a U-shaped conformation. The phosphate group engages in electrostatic interactions with the positively charged residues, $R110^{2.60}$, $R208^{ECL2}$, and $K210^{ECL2}$, but does not form direct interactions, such as hydrogen bonds. Moreover, the carboxylate of the serine moiety forms a direct salt bridge with $R286^{6.55}$ and hydrogen bonds with $Y135^{3.33}$ and $N309^{7.35}$. The amine group forms an electrostatic interaction with $E310^{7.36}$ and a hydrogen bond with the backbone carbonyl group of $F205^{ECL2}$. Of note, the oxygen atom in the *sn*-3 position forms a hydrogen bond with $N220^{5.40}$, whereas the ethoxy group in the *sn*-1 position has less contact with the receptor than the other moieties. Overall, these data suggest that GPR34 more firmly recognizes to the amine and carboxylate groups of the serine moiety rather than the phosphate group.

The hydrophobic pocket consists of a gap between the extracellular halves of TM4 and TM5 (TM4–5 gap; Fig. 2f). This gap is wider than those in the EDG family members of lipid receptors and the

phylogenetically related P2Y receptor (P2Y12), owing to different positions of TM4[27–31] (Supplementary Fig. 3a–g). However, a similarly wide gap is observed in the structure of the non-EDG LPA receptor LPA6[25] (Supplementary Fig. 3b, c). In GPR34, the gap is composed of hydrophobic residues in TM4 and TM5. Notably, TM5 contains the bulky residues $F219^{5.39}$ and $L223^{5.43}$, whereas the opposite position in TM4 has the small residues $A182^{4.53}$ and $G185^{4.56}$ (Fig. 2f), resulting in formation of an L-shaped hydrophobic pocket (Fig. 2d, f). The acyl chain of S3E-LysoPS is bent at the *cis*-9 double bond and fits along the L-shaped pocket. Consequently, the C1–C9 chain is exposed to the membrane environment, consistent with a previous study reporting that GPR34 is activated by LysoPS analogues attached to alkoxy amine chains with various hydrophobic tail lengths[32].

To validate the observed agonist interactions, we mutated receptor residues involved in S3E-LysoPS binding. Within the hydrophilic pocket, alanine mutants of the four aromatic residues $Y135^{3.33}$, $F205^{ECL2}$, $Y207^{ECL2}$, and $Y289^{6.58}$, which are critical for hydrophilic pocket formation, reduced potency of S3E-LysoPS ($pEC_{50}$) by over 100-fold (Fig. 2g, Supplementary Fig. 4a, b, and Supplementary Table 2). Moreover, $R286^{6.55}$A mutation abolishes agonist response, whereas alanine mutations of $R208^{ECL2}$, $N309^{7.35}$, and $E310^{7.36}$ reduced potency by ~10–30-fold. These observations are consistent with the fact that residues involved in head group recognition are highly conserved among vertebrates, indicating their functional importance for LysoPS receptors (Supplementary Fig. 5a, b). In contrast, alanine mutations in the hydrophobic pocket only reduced potency by up to 4-fold (Fig. 2g, Supplementary Fig. 4a, b, and Supplementary Table 2). These hydrophobic pocket residues are less highly conserved compared to those in the hydrophilic pocket, suggesting there is no strict spatial requirement to accommodate the acyl chain.

## Ligand access

The ligand-binding pocket of GPR34 is open toward both the membrane and extracellular space (Fig. 2c, d), suggesting that the ligand can enter the pocket laterally from the membrane and from the extracellular medium[32]. Unlike other lysophospholipids, such as LPA and S1P, which are both present in relatively high amounts as carrier-bound forms in extracellular fluids, LysoPS concentration in extracellular fluids is too low to activate receptors[33]. In addition, LysoPS is produced from PS by the extracellular enzyme PS-PLA1 in the outer leaflet of the plasma membrane[19], with no known pathways for production in the extracellular fluid. Interestingly, when added to the medium, recombinant PS-PLA1 protein activates GPR34 at the cellular level[34] (Supplementary Fig. 6a). Under these conditions, LysoPS is not present in the medium but, rather, is associated with cells (Supplementary Fig. 6b, c). When albumin, which can extract lysophospholipids from the membrane, is added simultaneously, PS-PLA1-induced GPR34 activation is dramatically weakened (Supplementary Fig. 6a, d), indicating that membrane-associated LysoPS, but not albumin-bound LysoPS, is capable of activating GPR34. Further, a PS-PLA1 S166A mutant, which has no enzyme activity, only weakly activates GPR34 (Supplementary Fig. 6a). These results, together with a ligand pocket open to the membrane, suggest that LysoPS enters the pocket laterally when produced on the outer leaflet of the plasma membrane by PS-PLA1. Moreover, albumin effectively inhibits M1-induced GPR34 activation in a dose-dependent manner (Supplementary Fig. 6e-g), suggesting lateral access of the synthetic GPR34 agonist, in addition to extracellular access.

To further investigate the lateral access, we generated three mutant GPR34 constructs, A182W, G185F, and G185W, which are designed to close the TM4-5 gap. The three mutants showed the same expression levels (Supplementary Fig. 6h) and responses to S3E-LysoPS (each 1 μM) similar to the wild-type GPR34 (Supplementary Fig. 6h). Among the mutants, the G185F mutant was activated by the recombinant PS-PLA1 protein to the same extent as the wild-type

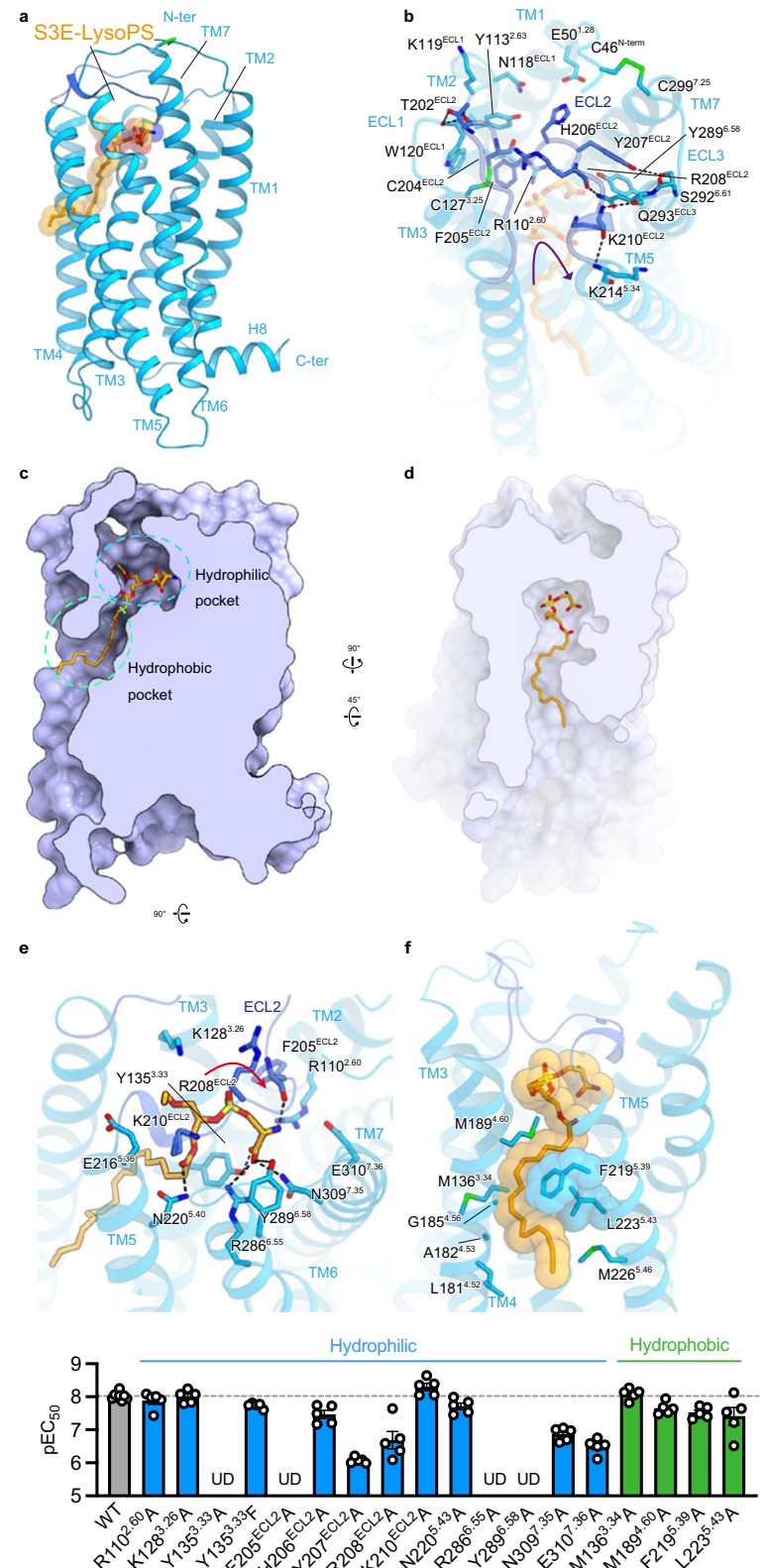

**Fig. 2 | S3E-LysoPS binding mode. a** Overall structure of the S3E-LysoPS-bound receptor. Disulfide bonds are shown as sticks. **b** Interactions between extracellular loop (ECL)2 and transmembrane helices (TMs). Black dashed lines indicate hydrogen-bonding interactions. **c, d** Cross-sectional views of the ligand-binding pocket, viewed from the membrane plane (**c**) and the extracellular side (**d**).

**e, f** Binding mode of S3E-LysoPS in the hydrophilic (**e**) and hydrophobic (**f**) pockets of the receptor. **g** Mutagenesis data for identifying the S3E-LysoPS interaction residues ($n = 8$ for the wild type and $n = 5$ for the mutants). Values are shown as the mean ± s.e.m. from at least three independent experiments performed in triplicate. Source data are provided as a Source Data file.

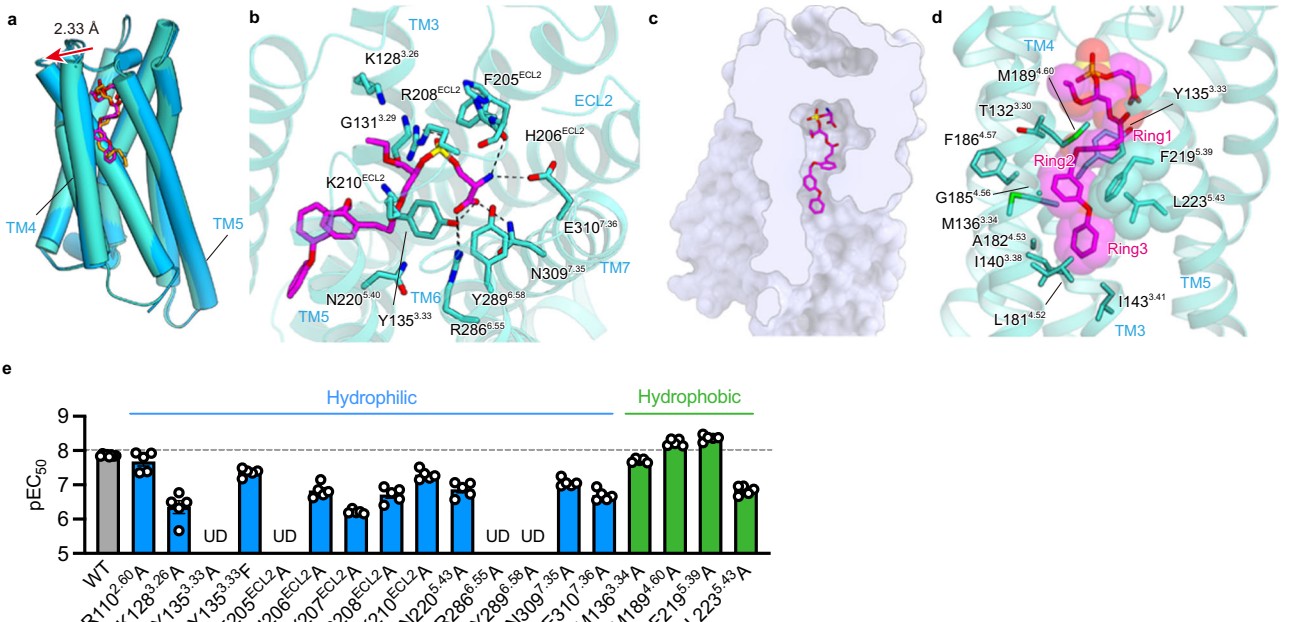

**Fig. 3 | M1 binding mode. a** Superimposition of the GPR34 structures bound to M1 (turquoise) and S3E-LysoPS (blue). The red arrow indicates the distance between the Cα atoms of L192[4.63] in the two structures. **b** Binding mode of M1 in the hydrophilic pocket. Black dashed lines indicate hydrogen-bonding interactions. **c** Cross-section of the ligand-binding pocket, viewed from the extracellular side.

**d** Binding mode of M1 in the hydrophobic pocket, highlighting the ring interactions. **e** Mutagenesis data for identifying the M1 interaction residues ($n = 8$ for the wild type and $n = 5$ for the mutants). Values are shown as the mean ± s.e.m. from at least three independent experiments performed in triplicate. Source data are provided as a Source Data file.

GPR34 (Supplementary Fig. 6i). By contrast, neither A182W nor G185W was activated by the recombinant PS-PLA$_1$ protein. These facts that G182W and G185W were activated by S3E-LysoPS, but not by PS-PLA$_1$, can be interpreted as follows: The bulky tryptophan side chain prevented the ligand's lateral access but not its access from outside of the cell. Thus, PS-PLA$_1$ may be the only LysoPS-producing enzyme that provides the ligand to GPR34. Accordingly, although LysoPS can access GPR34 from both the outer open space of the cell and laterally through the membrane, the LysoPS produced by PS-PLA$_1$ in the plasma membrane of GPR34-expressing cells migrates laterally in the plane of the plasma membrane to access GPR34.

## M1 binding mode

We next analysed the GPR34 structure bound to M1, the metabolically stable S3E-LysoPS analogue[22]. The overall structure of the M1-bound receptor superimposes well on the S3E-LysoPS-bound structure, with a root-mean-square deviation (RMSD) of 0.73 Å (Fig. 3a). M1 binds to the hydrophilic and hydrophobic pockets in a pose similar to that of S3E-LysoPS, and the head groups of M1 and S3E-LysoPS form comparable interactions with the hydrophilic pocket (Fig. 3b). Compared with the S3E-LysoPS-bound form, the extracellular portion of TM4 is displaced outwardly by 2.6 Å (Fig. 3a), due to key differences in the hydrophobic pockets. Here, the three aromatic rings of M1 (ring1, ring2, and ring3) are accommodated in the TM4–5 gap (Fig. 3c, d), with ring1 and ring3 oriented perpendicular to ring2. The bend between ring2 and ring3 fits along the L-shaped hydrophobic pocket, which superimposes with the position of the *cis*-9 double bond of 18:1 in S3E-LysoPS[17]. These aromatic moieties tightly interact with the receptor by stacking interactions with F219[5.39] and L223[5.43] (Fig. 3a). Overall, the larger opening of the TM4–5 gap accommodates the bulky aromatic groups of M1 well.

As for S3E-LysoPS, we mutated the residues involved in the M1 binding and observed overall effects is similar to those detected with S3E-lysoPS (Fig. 3e, Supplementary Fig. 4a, b, and Supplementary Table 2). However, within the hydrophobic pocket, F219[5.39]A mutation increased the potency of M1, suggesting that its bulkiness is not

essential for M1 binding. In contrast, L223[5.43]A mutation reduces potency by 10-fold (Fig. 3e), suggesting that the L-shaped hydrophobic constriction formed by L223[5.43] provides the necessary environment for binding M1, but not S3E-LysoPS.

## Validation of agonist binding modes by MD simulations

To validate the observed agonist binding modes, we performed 1-μs MD simulations of receptor–ligand complexes in a 1-palmitoyl-2-oleoyl-*sn*-glycero-3-phosphocholine (POPC) lipid bilayer environment. During the simulations, interactions between phosphoserines and receptors, observed in the cryo-EM structure are stably maintained for both *sn*-3 LysoPS derivatives: S3E-LysoPS and M1 (Fig. 4a, b, Supplementary Fig. 7a, b, Supplementary Movie 1, and Supplementary Discussion). The present results further suggest that anionic charge repulsion between the phosphate (PO$^-$) and serine CO$_2^-$ moieties could contribute to a preference for the U-shaped conformation of the hydrophilic portions of these ligands during the binding process. Metadynamics simulations support that the U-shaped conformation is the energy minimum in the energy landscape for both ligands (Supplementary Fig. 7c-e). This conformation may facilitate the stabilization of the interaction network between charged phosphoserine moieties (NH$_3^+$, CO$_2^-$, and PO$^-$) and the corresponding residues E310[7.36], N309[7.35], R286[6.55], and F205[ECL2] in the hydrophilic binding-site (Fig. 4a, b).

MD simulations also indicated that both S3E-LysoPS and M1 form significant stable interactions with the receptor (Fig. 4a, b), consistent with the mutagenesis analysis (Supplementary Discussion). Notably, the ligand RMSD (Root Mean Square Deviation) with respect to protein (the same below), illustrating the average change in displacement of ligand for a particular frame with respect to a reference frame (initial frame), is smaller in the M1-bound structure than the S3E-LysoPS-bound structure during the 1-μs MD (Supplementary Fig. 7f, g), indicating that the binding of M1 is more stable than that of S3E-LysoPS. Similarly, the ligand RMSF (Root Mean Square Fluctuation), showing the ligand's fluctuations broken down by atom, of M1 (both

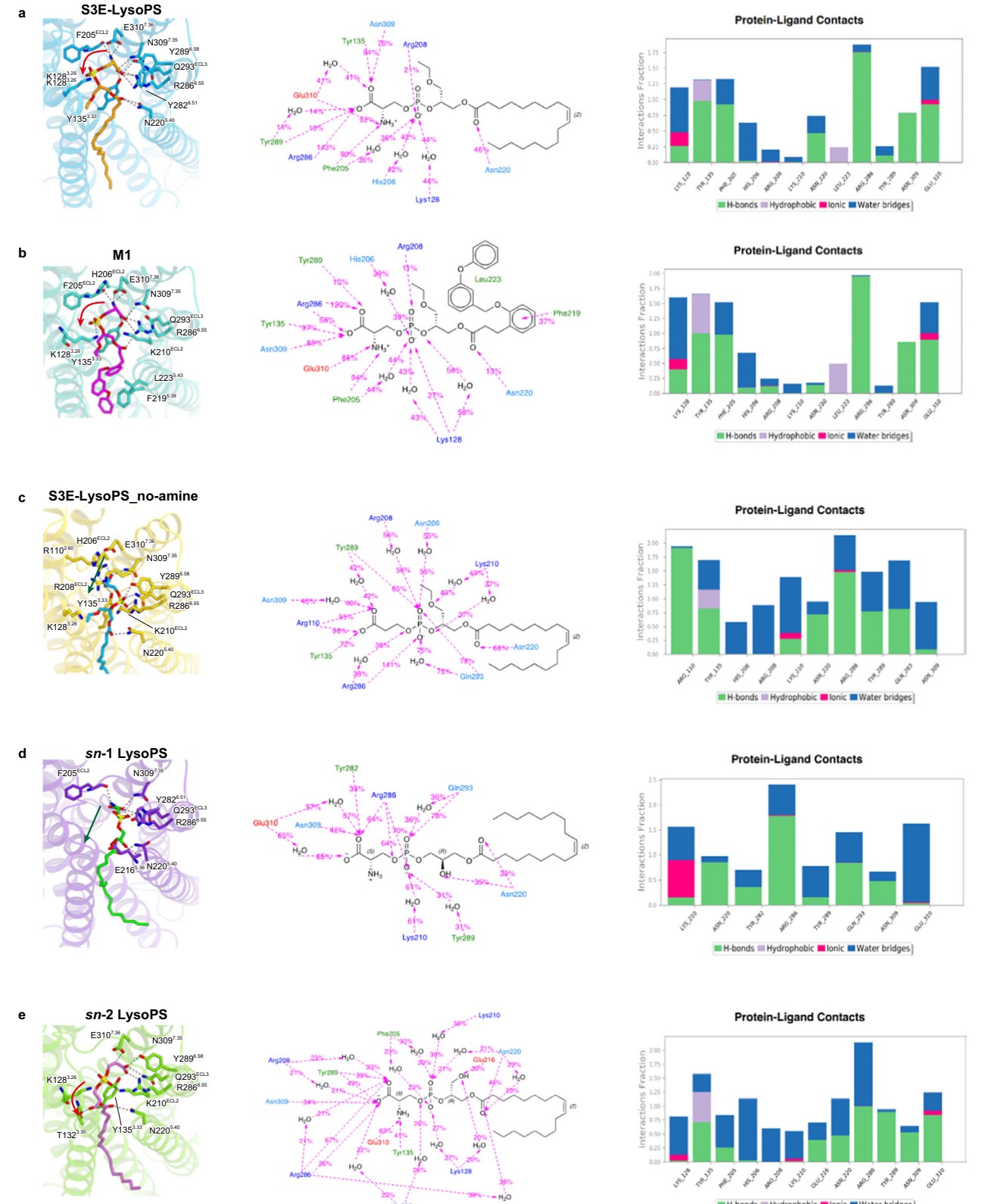

**Fig. 4 | Molecular dynamics (MD) simulations. a–e** Average structures and maintained ligand-protein (*P–L*) interactions during 1-μs MD simulations. The interactions were illustrated in percentage (percentage = number of frames_interaction present /number of frames_total). Middle panels: orange, negatively charged residues; blue, positively charged residues; cyan, polar residues; green, hydrophobic residues. Right panels: green, hydrogen bond; grey, hydrophobic interaction; magenta, ionic interaction; blue, salt bridge. In the cases of S3E-LysoPS (**a**) and M1 (**b**) bound GPR34 complexes, we showed the *P–L* interaction corresponding to mutagenesis results. In the cases of S3E-noamine, *sn*-1, and *sn*-2 LysoPS bound GPR34 complexes, only *P–L* interactions conserved over 50% were shown.

hydrophobic and hydrophilic parts at around 1 Å) is also lower as compared with S3E-LysoPS (hydrophobic part: 2 ~ 4 Å; hydrophilic part: around 1 Å).

The MD simulation with a ligand lacking the amine group in the serine head (S3E-LysoPS-no_amine) was also performed to validate the essentialness of the serine head group of the ligand (Fig. 4c). This calculation resulted in a binding mode switch from the U-shape of hydrophilic head to linear-shape within initial perturbation (Fig. 4c, Supplementary Fig. 7a, and Supplementary Movie 1). This is consistent with our previous findings regarding GPR34 ligand specificity; only LysoPS species with a phosphoserine head group can activate this receptor[15].

### Regioselectivity of LysoPS species binding to GPR34

Previous reports suggested that LysoPS is the endogenous ligand for GPR34[1,12]. Interestingly, GPR34 was more strongly activated by LysoPS with a fatty acid at the *sn*-2 position (*sn*-2-type LysoPS) than LysoPS with a fatty acid at the *sn*-1 position (*sn*-1-type LysoPS) (Supplementary Fig. 8a–d). Moreover, the LysoPS-producing enzyme PS-PLA$_1$ produces *sn*-2-type LysoPS, and thus the *sn*-2-type LysoPS appears be the endogenous ligand of GPR34. However, as both *sn*-1 and *sn*-2 LysoPS preparations are a mixture of *sn*-1- and *sn*-2-type LysoPS (Supplementary Fig. 8e–g), partially due to chemical *sn*-1 and *sn*-2 interconversion, it was unclear whether the *sn*-1-type LysoPS actually functions as a ligand for GPR34. Thus, it is difficult to obtain clear direct evidence with respect to the regioisomerism (that is, *sn*-1 and *sn*-2 formulae) of endogenous LysoPS. while the mutagenesis analysis of the entire ligand-binding pocket suggests that the hydrophilic pocket plays an essential role in the recognition of the polar head groups of both *sn*-1 and *sn*-2 type, as the synthetic ligands (Supplementary Fig. 9).

Accordingly, we performed docking simulations with natural *sn*-1 and *sn*-2 LysoPS species and compared the dynamics with those of the synthetic *sn*-3 analogue S3E-LysoPS. For *sn*-1 LysoPS (18:1), we found that the phosphoserine occupies a binding position with high probability that is distinct from the position common to S3E-LysoPS and M1 (Fig. 4d, Supplementary Fig. 7a, and Supplementary Movie 1). That is, the phosphoserine adopts a straight-line shape, similar to that of the deaminated derivative of S3E-LysoPS (Fig. 4c), rather than the U-shape observed in S3E-LysoPS and M1. Furthermore, in this case, instead of the carboxyl group (as in S3E-LysoPS or M1), the phosphate group primarily forms hydrogen bonds with K210$^{ECL2}$, R286$^{6.55}$, and Y289$^{6.58}$ (Fig. 4d). On the contrary, for *sn*-2 LysoPS (18:1), the docking pose is similar to that of S3E-LysoPS (Fig. 4a–e), in which the U-shaped conformation of the phosphoserine head group interacts with the hydrophilic residues, including E310$^{7.36}$, N309$^{7.35}$, R286$^{6.55}$, and F205$^{ECL2}$. This conformational mode is mostly maintained throughout the 1 μs MD simulation (Fig. 4e, Supplementary Fig. 7a, and Supplementary Movie 1). Critically, the above observation suggests that the U-shaped conformation of the hydrophilic head, common to S3E-LysoPS, M1, and *sn*-2 LysoPS (18:1), may represent the active form of the ligand, which essentially preserves the hydrophilic interaction network with the protein (E310$^{7.36}$, N309$^{7.35}$, R286$^{6.55}$, and backbone carbonyl of F205$^{ECL2}$). Thus, with respect to bioactive regioisomers of endogenous LysoPS, we assumed that the *sn*-2 isomer is bioactive while the *sn*-1 isomer is inactive.

### Receptor activation and G$_i$ coupling

Although the inactive GPR34 structure has not yet been solved, our active GPR34 structure provides mechanistic insight into receptor activation (Fig. 5a). In the homologous receptor P2Y12, positively charged residues, such as R256$^{6.55}$, form salt bridges with the phosphate groups of nucleic acids (Supplementary Fig. 10a), causing the 4-Å inward shift of TM6[27,35]. Likewise, S3E-LysoPS and M1 tightly interacts with R286$^{6.55}$ (Supplementary Fig. 10b, c), which is also observed in MD simulations of the *sn*-2 LysoPS-bound forms (Fig. 4d,

e). Below R286$^{6.55}$, Y282$^{6.51}$ hydrogen bonds with Y135$^{3.33}$, and H283$^{6.52}$ forms a π-stacking interaction with F227$^{5.47}$ (Fig. 5b). Analogous to P2Y12, interaction between LysoPS and R286$^{6.55}$ could induce an inward displacement of the extracellular portion of TM6, leading to formation of the central core interaction.

In most class A GPCRs, ligand binding rearranges hydrophobic contacts in the conserved P-I-F and CWxP motifs in the centres of TMs 3-5-6, leading to receptor activation[26,36]. P$^{5.50}$ is not conserved, as described above, and W$^{6.48}$ is replaced by F279$^{6.48}$ (Fig. 5b). However, these hydrophobic residues are tightly packed together with the nearby phenylalanine (Fig. 5b). A polar interaction network exists in the middle parts of TMs 1, 2, and 7, including the conserved D323$^{7.49}$ in the N/DPxxY motif and D100$^{2.50}$ (Fig. 5b). Formation of these interactions upon ligand binding creates an open cavity for TM5–6 on the intracellular side. Within this intracellular cavity, R152$^{3.50}$ and Y327$^{7.53}$ in the conserved DRY and N/DPxxY motifs are directed toward the centre of the transmembrane bundle (Fig. 5c), facilitating interactions with the C-terminal residues in the α5-helix of G$_i$ (Supplementary Discussion)[37]. The structures of the essential motifs and intracellular side are similar in the M1 and S3E-LysoPS-bound receptor (Supplementary Fig. 10d).

The cavity closely contacts the C-terminal α5-helix, and the cytoplasmic loops, particularly ICL2 and ICL3, may contribute to G protein interactions. Specifically, the α5-helix C-terminus forms a hydrogen bond with the backbone amide of S332$^{7.58}$ (Fig. 5d). Moreover, R152$^{3.50}$ hydrogen bonds with the backbone carbonyl of C351$^{G.H5.23}$ (superscript indicates the common Gα numbering [CGN] system), as typically observed in other GPCR-G$_i$ complexes[38]. The short ICL3 forms van der Waals interactions with the α5-helix and β-sheet of G$_i$, and N257$^{6.26}$ hydrogen bonds with E318$^{G.H4S6.12}$ and D341$^{G.H5.13}$ (Fig. 5e). The most characteristic feature of the GPR34-G-protein interface is the interaction at ICL2 (Fig. 5f), which is located at the root of the α5-helix, and thus has a significant effect on its orientation. In most receptor-G$_s$ and -G$_i$ complex structures[38–43], bulky hydrophobic residues in the α-helix of ICL2 fit into hydrophobic pockets formed by L194$^{G.S3.01}$, F336$^{G.H5.08}$, T340$^{G.H5.12}$, and I343$^{G.H5.15}$ in the Gα$_i$ subunits (Fig. 5g). In some cases (i.e., LPA$_1$-G$_i$ complex)[29,30], ICL2 adopts a disordered conformation, but M153$^{ICL2}$ still fits into the hydrophobic pocket of G$_i$ (Fig. 5h). Conversely, in GPR34, the hydrophobic residues in ICL2 do not fit in the pocket (Fig. 5f). Rather, I160$^{ICL2}$ and Q161$^{ICL2}$ form superficial interactions with the αN, α5-helix, and β-sheet of G$_i$. Due to these differences, the α5-helix in the GPR34-G$_i$ complex is 10° perpendicular to the receptor, compared to its position in the LPA$_1$-G$_i$ complex (Fig. 5i). Overall, the GPR34-G$_i$ coupling interaction is characteristic relative to that of other GPCR-G$_i$ complexes and extends the diverse binding modes observed for G$_i$ compared to G$_s$.

## Discussion

In summary, the S3E-LysoPS-bound cryo-EM structure revealed that the acyl chain is accommodated in the TM4-TM5 gap (Fig. 2f), a characteristic shared among some types of lipid-sensing GPCRs such as LPA$_6$. MD simulations based on the cryo-EM structure further showed that the hydrophilic head groups of *sn*-2 and *sn*-3 LysoPS can adopt U-shapes and form tight interactions with charged residues in the hydrophilic receptor pocket (Fig. 4a–e). Notably, the amine and carboxylate of the serine moiety are tightly recognized by E310$^{7.36}$ and R286$^{6.55}$ (Fig. 4a), respectively, consistent with the serine-specific recognition by the LysoPS receptor GPR34. In contrast, the head group of *sn*-1 LysoPS can only adopt a straight shape and does not form the aforementioned stable interactions (Fig. 4d), raising the possibility that *sn*-1 LysoPS may not be the true active species for GPR34, and *sn*-2 LysoPS is the genuine ligand for GPR34. The fact that PS-PLA$_1$, which produces *sn*-2 LysoPS, activated GPR34 at the cellular level[1] also strongly reinforces this hypothesis. Together with the results of our cell-based assay (Supplementary Fig. 8a–d) and previous reports, this

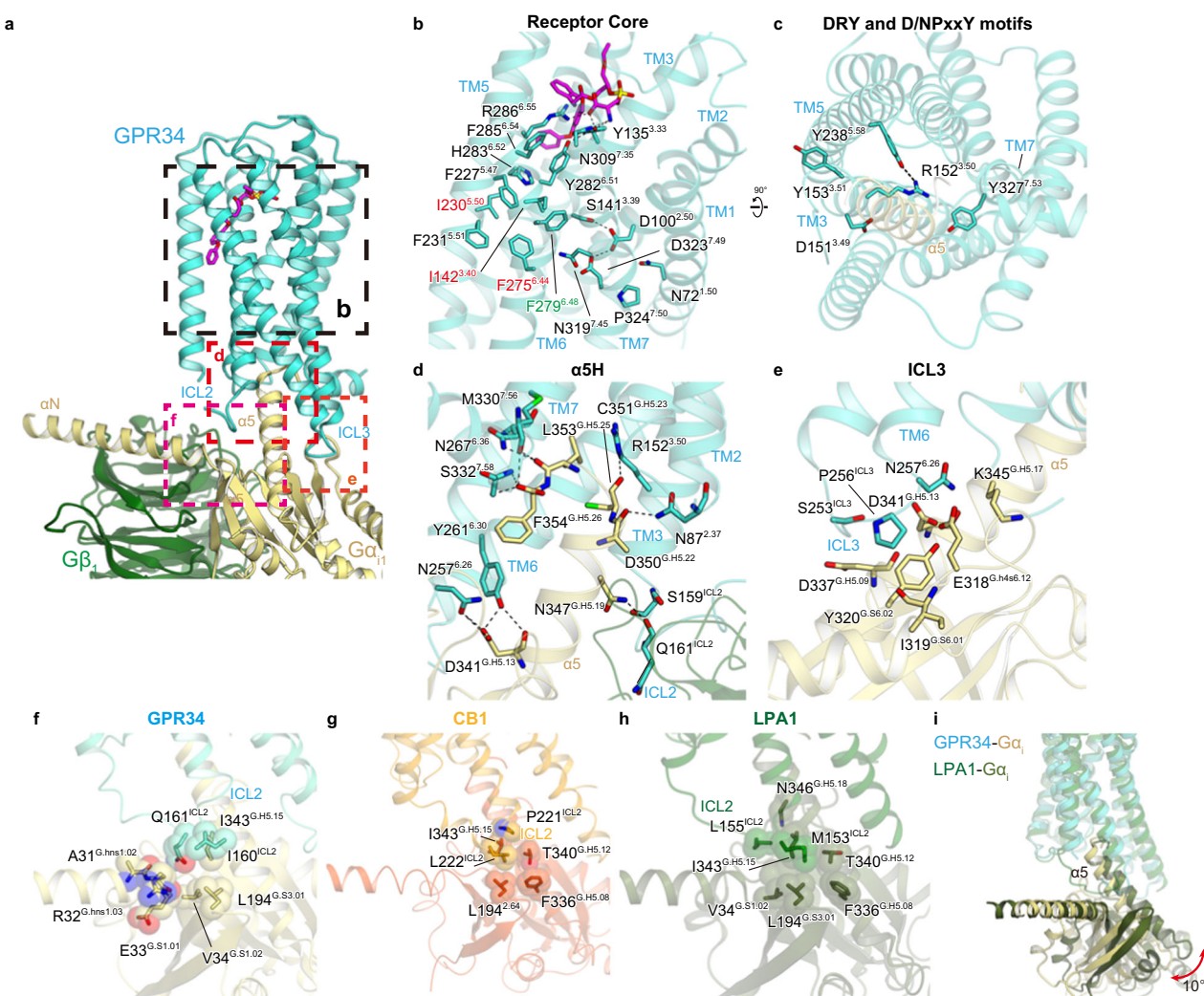

**Fig. 5 | G-protein activation. a** G-protein interface. **b** Interactions at the receptor core. **c** DRY and N/DPxxY motifs. Black dashed lines indicate hydrogen bonds. **d** Main hydrogen-bonding interactions between the receptor and the α5-helix of Gα$_{i1}$. **e** Interactions between intracellular loop (ICL)3 and G$_i$. **f–h** Interactions between ICL2 and G$_i$ in GPR34 (**f**), CB1 (PDB 6N4B) (**g**), and lysophosphatidic acid (LPA)$_1$ (PDB 6N4B) (**h**). **i** Comparison of the G-protein positions in the GPR34 and LPA$_1$ (PDB 7YU4) structures.

structural study supports the postulate that a special form of LysoPS; i.e., *sn*-2 LysoPS, is the physiological ligand of GPR34. This is one possible reason for the controversy regarding whether LysoPS can activate GPR34[11], since only *sn*-1 LysoPS was tested in that study. Critically, the metabolically stable agonist M1 nicely fits into the L-shaped hydrophobic pocket in the TM4–5 gap, forming quite stable hydrophilic interactions with the receptor. Overall, we anticipate that detailed SAR information and physiological functional data in the present and future studies will enable to access to therapeutic targeting of GPR34.

The binding mode of the acyl chain in GPR34 differs substantially from that in other lysophospholipid receptors (Supplementary Fig. 3h–m). For example, EDG receptors accommodate the acyl chain within a transmembrane pocket (Supplementary Fig. 3i–l), whereas GPR34 does so in the TM4–5 gap. However, the crystal structure of the non-EDG LPA receptor LPA$_6$ suggests that the TM4-5 gap can also accommodate the acyl chain (Supplementary Fig. 3c). Because both GPR34 and the non-EDG family LPA receptors are homologous to P2Y receptors, this suggests that acyl chain accommodation in the TM4–5 gap is a common feature of P2Y-like lysophospholipid receptors. It is interesting to consider why EDG and P2Y-like lysophospholipid receptors accommodate acyl chains differently. P2Y receptors are a family of purinergic G protein-coupled receptors activated by

nucleotides[3], such as adenosine triphosphate. P2Y-like lysophospholipid receptors evolved from P2Y family members to receive the acyl chain linked to the phosphate head. Notably, compared to EDG receptors, the phosphate-binding site of P2Y receptors is buried inside the transmembrane bundle (Supplementary Fig. 3m). We therefore propose that due to limited space, P2Y receptors used the membrane-facing hydrophobic region to evolve as lipid receptors.

The LysoPS receptors P2Y10 and GPR174 share 50% sequence identity, while GPR34 shows a more distant relationship. There is no evident conservation of residues necessary for LysoPS binding (Supplementary Fig. 11). After the submission of this manuscript, the GPR174 structure bound to the endogenous agonist *sn*-1 LysoPS 18:1 was reported[44] (PDB 7XV3). Together with the AlphaFold-2 (AF2)[45]-predicted P2Y10 structure, we performed a structural comparison of the three LysoPS receptors (Fig. 6a). In GPR174, as well as GPR34, the polar head group and acyl chain of the ligand are accommodated within the hydrophilic and hydrophobic pockets (Fig. 6b, c). Both receptors share a similar conformation in which the polar heads of the ligands are curved and aligned with each other. However, the orientations and relative positions differ significantly between GPR174 and GPR34. Notably, GPR174 possesses multiple positively charged residues that directly recognize the phosphate group, in contrast to the cryo-EM structure of GPR34. Interestingly, the polar residues

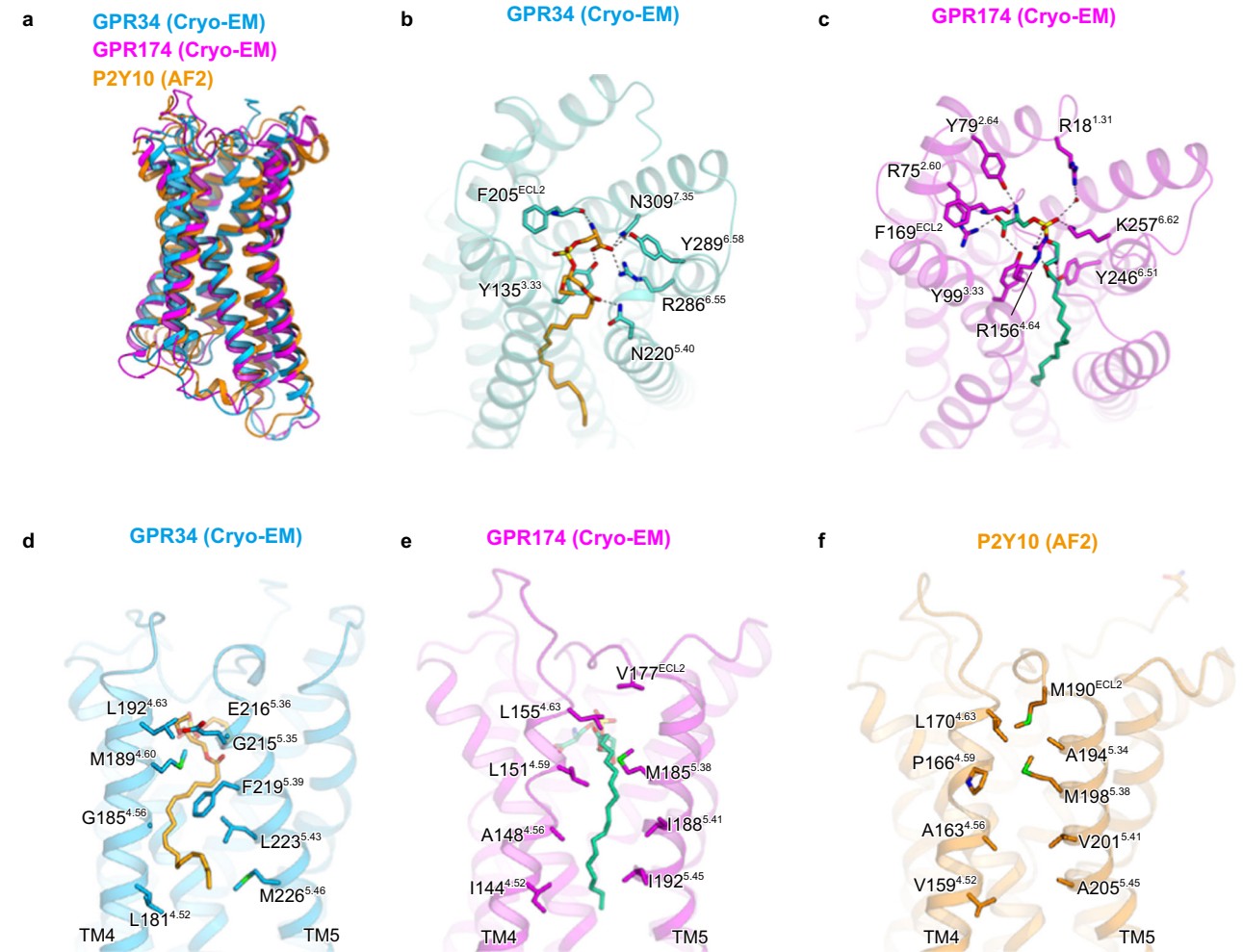

**Fig. 6 | Structural comparison of GPR34, GPR174, and P2Y10. a** Superimposition of the structures of GPR34, GPR174, and AF2-predicted P2Y10. **b, c** Comparison of the interactions with the polar head group in GPR34 (**b**) and GPR174 (**c**). **d–f** Comparison of the TM4-5 gaps in GPR34 (**d**), GPR174 (**e**), and AF2-predicted P2Y10 (**f**).

recognize both the amine and carboxylate groups of the serine moieties in a similar manner. Despite the disparities in the binding positions of the ligands, it is noteworthy that the conserved Y135/Y99 and F205/F169 residues are involved in serine recognition. In the hydrophobic pocket within the TM4-5 gap, the residues F219[5.39] and L223[5.43], which form the characteristic L-shaped pocket in GPR34, are not conserved in P2Y10 and GPR174 (Supplementary Fig. 11). Consequently, the TM4–5 gaps in these receptors adopt more open conformations (Fig. 6d–f). In GPR174, the acyl chain assumes a straight conformation within the gap. This structural comparison revealed that the L-shaped pocket in the TM4-5 gap is a distinct feature specific to GPR34. Earlier studies on structure-activity relationships have demonstrated that modifications of the acyl chain can alter ligand selectivity for LysoPS receptors. Just like the recognition of the aromatic rings of M1, the recognition of the L-shaped pocket is crucial for GPR34 selectivity.

## Methods

### Preparation of *sn*-1 LysoPS (18:1) and *sn*-2 LysoPS (18:1)

The *sn*-1 LysoPS (18:1) and *sn*-2 LysoPS (18:1) agonists were prepared as described previously[46]. Briefly, di-oleoyl (18:1) phosphatidylserine (PS) (di-18:1-PS) from Avanti Polar Lipids (Alabaster, AL, USA) was digested with *Rhizomucor miehei* lipase, which has intrinsic phospholipase $A_1$ (PLA$_1$) activity. The resulting *sn*-2 LysoPS (18:1) was stabilized by bringing the solvent to a mildly acidic pH of 4.0 to prevent the acyl migration

reaction. The PLA$_1$ reaction mixture containing *sn*-2 LysoPS (18:1), di-18:1-PS, and oleic acid was then subjected to C18-based reverse-phase cartridge column chromatography to separate *sn*-2 LysoPS (18:1). After obtaining a pure *sn*-2 LysoPS (18:1) preparation, the solvent was changed to alkaline conditions (pH 9.0) to facilitate the acyl migration reaction. After neutralization, the resulting LysoPS was used as *sn*-1 LysoPS (18:1). We assessed the purities of *sn*-2 LysoPS (18:1) and *sn*-1 LysoPS (18:1) preparations by liquid chromatography–tandem mass spectrometry (LC-MS/MS) and confirmed that they were >90% pure.

### Transforming growth factor (TGF)-α shedding assay

The TGF-α shedding assay was performed as described previously[47]. Briefly, HEK293A cells were seeded in 6-well plates at a density of $4 \times 10^5$ cells/well and cultured for 1 day in a 5% $CO_2$ incubator at 37 °C. Cells were then transfected with a mixture of plasmids encoding alkaline phosphatase-tagged (AP)-TGFα (500 ng), human GPR34 (200 ng), and $G\alpha_{q/i1}$, a chimeric Gα protein (100 ng), using polyethyleneimine (Polysciences, Inc., Warrington, PA, USA), and cultured for an additional day. Negative control cells were transfected with empty plasmid instead of the GPR34-encoding plasmid. Transfected HEK293A cells were harvested with 0.05% trypsin/EDTA and seeded in 96-well plates ($2.5 \times 10^4$ cells/well). Cells were then treated with various LysoPS agonists and a PS-PLA$_1$ recombinant protein in the presence of Ki16425, a lysophosphatidic acid (LPA)$_{1/3}$ antagonist (final concentration, 3 mM) in 0.01% bovine serum albumin (BSA)/HBSS, for 60 min at 37 °C. After

centrifugation, the supernatant was transferred to another plate and 10 mM *p*-NPP was added to both the supernatant and cell plates, at a volume of 80 mL/well. Finally, the optical density at 405 nm ($OD_{405}$) was measured with a SpectraMAX ABS Plus (Molecular Devices, San Jose, CA, USA), before and after incubation at room temperature. AP-TGFα release was calculated as follows:

$$AP - TGF\alpha \ release(\%) = DOD405_{Sup}/(DOD405_{Sup} + DOD405_{Cell}) \times 100 \times 1.25.$$

In this equation, we multiply by 1.25 to convert the amount of AP-TGFα in the transferred supernatant (80 mL) to the amount of AP-TGFα in total supernatant (100 mL). We then calculated GPCR activation as:

$$GPCR \ activation(\%) = AP - TGF\alpha \ release \ under \ stimulated.$$
$$conditions(\%) - AP - TGF\alpha \ release \ under \ unstimulated \ conditions(\%)$$

GPCR activation levels were fit to four-parameter sigmoidal concentration–response curves, using Prism9 software (GraphPad, San Diego, CA, USA), and $pEC_{50}$ and $E_{max}$ values were obtained from the curves.

## cAMP assay

The cAMP assay was performed with GloSensor cAMP Biosensor (Promega, Madison, WI, USA), as previously described[47]. Briefly, HEK293A cells were seeded and cultured as described for the TGF-α shedding assay above. Cells were then transfected with a mixture of plasmids encoding GloSensor-22F (1 mg) and human GPR34 (200 ng), using polyethyleneimine (Polysciences), and cultured for an additional day. Negative control cells were transfected with empty plasmid instead of the GPR34-coding plasmid. Transfected HEK293A cells were harvested in Dulbecco's Phosphate-Buffered Saline (D-PBS), containing 2 mM EDTA, and resuspended in 0.01% BSA/HBSS. Cells were then seeded in half-area white 96-well plates ($3.5 \times 10^4$ cells/well) and loaded with D-Luciferin (final concentration, 2 mM). After incubation in the dark for 2 h at room temperature, basal luminescence was measured by a SpectraMAX L microplate reader (Molecular Devices). Cells were then treated with forskolin (final concentration, 10 mM) and various LysoPS in the presence of Ki16425, an $LPA_{1/3}$ antagonist (final concentration, 3 mM) in 0.01% BSA/HBSS, and post-stimulus luminescence was kinetically measured for 20 min at room temperature. We then calculated cAMP (% Forskolin stimulation) as follows: post-stimulus luminescence was normalized by dividing the raw values by the basal luminescence, and normalized luminescence in both agonist and forskolin-treated conditions was divided by that in forskolin-only treated conditions. To obtain $pEC_{50}$ and $E_{max}$ values, cAMP signals were fitted to four-parameter sigmoidal concentration–response curves, using Prism9 software (GraphPad).

## GPR34 activation by PS-PLA₁

Recombinant $PS\text{-}PLA_1$ was prepared as described previously[48], with minor modifications. In brief, HEK293A cells were transfected with the plasmid (1 mg) encoding wild-type (WT) or S166A-mutant mouse PS-$PLA_1$, using Lipofectamine 2000 (Thermo Fisher Scientific, Waltham, MA, USA). Negative control cells were transfected with the empty plasmid. After 4 h, the medium was changed to Opti-MEM, and the HEK293A cells were cultured for 72 h in a 5% $CO_2$ incubator at 37 °C. The culture supernatant was then collected and centrifuged at 400 × g for 5 min, and the resulting supernatant was used as recombinant $PS\text{-}PLA_1$. For the evaluation of GPR34 activation by $PS\text{-}PLA_1$, the TGFα shedding assay was performed, as described above, using the recombinant $PS\text{-}PLA_1$ protein in place of LysoPS.

## Sample preparation for LC-MS/MS analysis

The amount of LysoPS in HEK293A cells and in supernatant from HEK293A cells stimulated by recombinant $PS\text{-}PLA_1$ was determined by LC-MS/MS analysis. Samples for LC-MS/MS analysis were prepared as described previously[49]. Briefly, HEK293A cells were stimulated by PS-$PLA_1$ as described above, and the entire supernatant was collected. Cells were treated with ice-cold acidic MeOH, containing 100 nM 17:0-LPA, and incubated for 10 min at room temperature; LysoPS dissolved in MeOH was then collected. For supernatant samples, 10 mL of the collected supernatant was added to 90 mL of acidic MeOH, containing 111 nM 17:0-LPA. Both cell and supernatant samples were passed through a filter with a 0.2 mM pore size and a 4 mm inner diameter and subjected to LC-MS/MS analysis, as described below.

## LC-MS/MS analysis

LC-MS/MS analysis was performed as described previously[49], using an LC-MS/MS system consisting of a Vanquish HPLC system and a TSQ Altis™ Triple-Stage Quadrupole Mass Spectrometer (Thermo Fisher Scientific). For HPLC, samples were separated in the L-column2 (100 mm × 2 mm, 3 mm particle size, CERI), using a gradient solution consisting of solvent A (5 mM ammonium formate in water, pH 4.0) and solvent B (5 mM ammonium formate in acetonitrile, pH 4.0) at 200 ml/min. LysoPS was then monitored in the negative ion mode, using MS/MS. At MS1, the m/z values of $[M + H]^+$ ion for LysoPS were selected. At MS3, lysophosphatidic acid fragments derived from LysoPS were detected. The amount of LysoPS in samples was calculated based on the standard curve of 18:1-LysoPS.

## Preparation of anti-GPR34 serum

Anti-GPR34 serum was obtained by performing DNA immunization[50] of *Gpr34*-knockout (KO) mice (C57BL/6 J background) to ensure the immunogenicity of GPR34. Male 7-weeks old, specific pathogen-free *Gpr34*-KO mice ($n = 5$) were intramuscularly injected with pCAGGS (100 mg) plasmid, which encodes a mouse GPR34–GroEL fusion protein, and electroporated in vivo. Immunization was performed five times in total, once every 2 weeks. Two days after the last immunization, mice were boosted by intrasplenic administration of mouse GPR34-expressing HEK293T cells ($2 \times 10^7$ cells), and serum was collected 3 days later. This serum was used as anti-GPR34 serum. During immunization, mice were housed in climate-controlled (23 °C) facilities with a 12-h light/12-h dark cycle. The animal experiment was approved by the animal ethics committee of the University of Tokyo prior to their commencement and performed in accordance with approved protocols.

## Evaluation of GPR34 mutant expression

Expression of GPR34 mutants was measured by flow cytometry. In brief, HEK293A cells were transfected with human GPR34-encoding plasmid (250 ng), using polyethyleneimine. Cells were then suspended in 200 ml of D-PBS, containing 2 mM EDTA, and dispensed into 96-well V-bottom plates. After centrifugation for 1 min at 700 × g, the cells were suspended in 200 ml/well of FACS buffer (D-PBS, containing 0.5% BSA and 2 mM EDTA) and incubated for 30 min on ice. Cells were then centrifuged again for 1 min at 700 × g, resuspended in 25 ml/well of anti-human GPR34 serum (1/100 diluted), and incubated for 30 min on ice. After centrifugation for 1 min at 700 × g, cells were washed with D-PBS, resuspended in 25 ml/well of goat anti-mouse IgG conjugated with Alexa488 (Thermo Fisher Scientific, 10 mg/ml), and incubated for 15 min on ice. Cells were then centrifuged a final time for 1 min at 700 × g, washed with D-PBS, and resuspended in 150 ml/well of D-PBS, containing 2 mM EDTA. Flow cytometry analysis was performed with the BD FACSLyric Flow Cytometry System (BD Biosciences, Franklin Lakes, NJ, USA), and the data were analysed by FlowJo Software (FlowJo, Ashland, OR, USA).

## Expression and purification of human GPR34

GPR34 was subcloned into a modified pEG Bacmam vector[51], with an N-terminal haemagglutinin signal peptide, followed by the Flag-tag

epitope (DYKDDDD), and a C-terminal tobacco etch virus (TEV) protease recognition site, followed by an enhanced green-fluorescent protein (EGFP)-His tag[52]. Recombinant baculovirus was prepared using the Bac-to-Bac baculovirus expression system and *Spodoptera frugiperda* Sf9 insect cells (Thermo Fischer Scientific). The receptor was expressed in HEK293S GnTI- (N-acetylglucosaminyl-transferase I-negative) cells, obtained from the American Type Culture Collection (ATCC; catalogue no. CRL−3022).

To purify the S3E-LysoPS-bound receptor, harvested cells were solubilized in buffer, containing 20 mM Tris-HCl, pH 8.0, 150 mM NaCl, 1% lauryl maltose neopentyl glycol (LMNG; Anatrace, Maumee, OH, USA), 0.1% cholesteryl hemisuccinate (CHS), 10% glycerol, and 10 μM S3E-LysoPS, for 1 h at 4 °C. The supernatant was separated from the insoluble material by ultracentrifugation at 180,000 × g for 30 min and then incubated with anti-Flag-M1 resin (Merck & Co., Inc., Rahway, NJ, USA,) for 2 h. Bound resin was washed with 20 column volumes of buffer, containing 20 mM Tris-HCl, pH 8.0, 500 mM NaCl, 0.05% glyco-diosgenin (GDN; Anatrace), 1 μM S3E-LysoPS, 10% glycerol, and 5 mM CaCl$_2$. The receptor was then eluted in 20 mM Tris-HCl, pH 8.0, 150 mM NaCl, 0.1% GDN, 1 μM S3E-LysoPS, 10% glycerol, 5 mM EDTA, and 0.15 mg ml$^{-1}$ Flag peptide. The receptor was concentrated and loaded onto a Superdex 200 10/300 column in 20 mM Tris-HCl, pH 8.0, 150 mM NaCl, 0.01% GDN, and 1 μM agonist, and peak fractions were pooled and frozen in liquid nitrogen.

To purify the M1-bound receptor, harvested cells were disrupted by sonication in buffer containing 20 mM Tris-HCl, pH 8.0, 200 mM NaCl, and 10% glycerol. The crude membrane fraction was collected by ultracentrifugation at 180,000 × g for 1 h, and the membrane fraction was solubilized in buffer, containing 20 mM Tris-HCl, pH 8.0, 200 mM NaCl, 2% LMNG (Anatrace), 0.4% CHS, and 10 μM M1, for 1 h at 4 °C. The supernatant was separated from the insoluble material by ultracentrifugation at 180,000 × g for 20 min and incubated with TALON resin (Clontech, Mountain View, CA, USA) for 30 min. Bound resin was washed with ten column volumes of buffer, containing 20 mM Tris-HCl, pH 8.0, 500 mM NaCl, 0.05% GDN, 1 μM M1, and 15 mM imidazole. The receptor was then eluted in buffer, containing 20 mM Tris-HCl, pH 8.0, 500 mM NaCl, 0.05% GDN, 1 μM M1, and 200 mM imidazole. The receptor was concentrated and loaded onto a Superdex 200 10/300 Increase size-exclusion column, equilibrated in buffer containing 20 mM Tris-HCl, pH 8.0, 150 mM NaCl, 0.01% GDN, and 1 μM M1. Peak fractions were pooled and frozen in liquid nitrogen.

### Expression and purification of the G$_i$ heterotrimer
The G$_i$ heterotrimer was expressed and purified using the Bac-to-Bac baculovirus expression system, according to the method reported previously[38]. In brief, Sf9 insect cells were infected at a density of 3-4 × 10$^6$ cells ml$^{-1}$ with a 100th volume of two viruses, one encoding the WT human Gα$_{i1}$ subunit and the other encoding the WT bovine Gγ$_2$ subunit and the WT rat Gβ$_1$ subunit containing a His$_8$ tag followed by an N-terminal tobacco etch virus (TEV) protease cleavage site. Infected Sf9 cells were incubated in Sf900II medium at 27 °C for 48 h and collected by centrifugation at 6,200 × g for 10 min. The collected cells were then lysed in buffer containing 20 mM Tris, pH 8.0, 150 mM NaCl, and 10% glycerol. The Gα$_{i1}$β$_1$γ$_2$ heterotrimer was solubilized at 4 °C for 1 h in buffer containing 20 mM Tris, pH 8.0, 150 mM NaCl, 10% glycerol, 1% (w/v) *n*-dodecyl-beta-D-maltopyranoside (DDM; Anatrace), 50 μM GDP (Roche), and 10 mM imidazole. The soluble fraction containing G$_{i1}$ heterotrimers was then isolated by ultracentrifugation at 186,000 × g for 20 min, and the supernatant was mixed with Ni-NTA Superflow resin (QIAGEN, Hilden, Germany) and stirred at 4 °C for 1 h. Bound resin was washed with 10 column volumes of buffer, containing 20 mM Tris pH 8.0, 150 mM NaCl, 0.02% DDM, 10% glycerol, 10 μM GDP, and 30 mM imidazole. G$_{i1}$ heterotrimers were then eluted with two column volumes of buffer, containing 20 mM Tris, pH 8.0, 150 mM NaCl, 0.02% (w/v) DDM, 10% (v/v) glycerol, 10 μM GDP and 300 mM imidazole. The

eluted fraction was dialysed overnight at 4 °C against 20 mM Tris, pH 8.0, 50 mM NaCl, 0.02% DDM, 10% glycerol, and 10 μM GDP. To cleave the histidine tag, TEV protease was added during the dialysis. The dialysed fraction was then incubated again with Ni-NTA Superflow resin at 4 °C for 1 h. The flow-through was collected and purified by ion-exchange chromatography on a HiTrap Q HP column (GE Healthcare Life Sciences, Chicago, IL, USA), using Buffer I1 (20 mM Tris, pH 8.0, 50 mM NaCl, 0.02% DDM, 10% glycerol, and 1 μM GDP) and Buffer I2 (20 mM Tris, pH 8.0, 1 M NaCl, 0.02% DDM, 10% glycerol, and 1 μM GDP).

### Expression and purification of scFv16
The gene encoding scFv16 was synthesized (GeneArt, Regensburg, Germany) and subcloned into a modified pFastBac vector, with the resulting construct encoding the GP67 secretion signal sequence at the N-terminus, and a His$_8$ tag, followed by a TEV cleavage site at the C-terminus[38]. His$_8$-tagged scFv16 was expressed and secreted by Sf9 insect cells, as previously reported[38]. Sf9 cells were collected by centrifugation at 5000 × g for 10 min, and the secreta-containing supernatant was combined with 5 mM CaCl$_2$, 1 mM NiCl$_2$, 20 mM HEPES, pH 8.0, and 150 mM NaCl. The supernatant was mixed with Ni Sepharose excel (Cytiva) and stirred for 1 h at 4 °C. The bound resin was washed with buffer containing 20 mM HEPES, pH 8.0, 500 mM NaCl, and 20 mM imidazole, and further washed with 10 column volumes of buffer containing 20 mM HEPES, pH 8.0, 500 mM NaCl, and 20 mM imidazole. The protein was then eluted with 20 mM Tris, pH 8.0, 500 mM NaCl, and 400 mM imidazole, and the eluted fraction was concentrated and loaded onto a Superdex 200 10/300 Increase size-exclusion column, equilibrated in buffer containing 20 mM Tris (pH 8.0) and 150 mM NaCl. Peak fractions were pooled, concentrated to 5 mg ml$^{-1}$ with a centrifugal filter device (10-kDa MW cut-off; MilliporeSigma, Burlington, MA, USA), and frozen in liquid nitrogen.

### Formation and purification of the GPR34-G$_i$ complex
Purified GPR34-GFP was mixed with a 1.2 molar excess of G$_i$ heterotrimer, scFv16, and TEV protease. After the addition of apyrase (to catalyse hydrolysis of unbound GDP) and agonist (final concentration, 10 μM), the coupling reaction was performed overnight at 4 °C. To remove excess G protein, the complexing mixture was purified by M1 anti-Flag affinity chromatography. Bound complex was washed in buffer, containing 20 mM Tris HCl, pH 8.0, 150 mM NaCl, 0.01% GDN, 1 μM agonist, 10% glycerol, and 5 mM CaCl$_2$. The complex was then eluted in 20 mM Tris-HCl, pH 8.0, 150 mM NaCl, 0.01% GDN, 10 μM agonist, 10% glycerol, 5 mM EDTA, and Flag peptide. The GPR34-G$_i$-scFv16 complex was purified by size exclusion chromatography on a Superdex 200 10/300 column in 20 mM Tris-HCl, pH 8.0, 150 mM NaCl, 0.01% GDN, and 1 μM agonist, and peak fractions were concentrated to -12 mg ml$^{-1}$ for electron microscopy studies.

### Cryo-EM grid preparation and data collection
For cryo-EM grid preparation of GPR34-G$_i$ complexes, 3 μl of protein at a concentration of -10 mg ml$^{-1}$ were loaded onto glow-discharged holey carbon grids (Quantifoil Au 300 mesh R1.2/1.3 or Quantifoil Cu/Rh 300 mesh R1.2/1.3), after which these were plunge-frozen in liquid ethane, using a Vitrobot Mark IV (Thermo Fischer Scientific). Cryo-EM imaging was collected on a Titan Krios at 300 kV, using a Gatan K3 Summit detector. Images were obtained at a dose rate of about 8.0 e − /Å2 s − 1, with a defocus ranging from −1.2 to −2.2 μm, using SerialEM software[53]. Total exposure time was 8 s, with 40 frames recorded per micrograph. A total of 2,358 and 2,674 movies were collected for S3E-LysoPS- and M1-bound GPR34-G$_i$ complexes, respectively.

### Image processing
For the S3E-LysoPS−GPR34-G$_{ii}$ complex, single-particle analysis of GPR34-G$_i$ complexes was performed with RELION-3.1[54,55]. Dose-

fractionated image stacks were subjected to motion correction by MotionCorr2[56], and contrast transfer function (CTF) parameters for micrographs were estimated by CTFFIND-4.0[57]. 2,012,061 particles were extracted, and the initial model was generated in RELION 3.1. Particles were subjected to several rounds of two-dimensional (2D) and three-dimensional (3D) classifications, resulting in the optimal classes of particles and yielding 258,700 particles. Particles were next subjected to 3D refinement, CTF refinement, and Bayesian polishing[58]. Following 3D refinement, particles were further classified into four classes, without alignment, using a mask covering the receptor. The 109,160 particles in the best class were subjected to 3D refinement and then further classified into three classes, without alignment, using a mask covering the extracellular half of the receptor. The 79,925 particles in the best class were subjected to 3D refinement, and post-processing, and cryoSPARCv4.0[59] nonuniform refinement, yielded a map having a nominal overall resolution of 3.3 Å, with the gold standard Fourier Shell Correlation (FSC = 0.143) criteria[60]. The 3D model was locally refined with a mask on the receptor, and as a result, the receptor has a higher nominal resolution of 3.4 Å. The processing strategy is described in Supplementary Fig. 1.

For the M1-bound GPR34-$G_i$ complex, all acquired movies were binned by 2× and were dose-fractionated and subjected to beam-induced motion correction implemented in RELION-4.0[61]. The contrast transfer function (CTF) parameters were estimated using patch CTF estimation in cryoSPARCv4.0. 1,398,266 particles were picked up by TOPAZ based auto-picking[62] and extracted. After the 3D classification and non-uniform refinement, 460,240 particles were subjected to RELION-4.0 and further classified without alignment, using a mask covering the receptor. The 236,096 particles in the best class were subjected to 3D refinement, CTF refinement, and Bayesian polishing. Then, the particles were subjected to cryoSPARCv4.0 and non-uniform refinement, yielded a map having a nominal overall resolution of 2.8 Å, with the gold standard Fourier Shell Correlation (FSC = 0.143) criteria. The 3D model was locally refined with a mask on the receptor, and as a result, the receptor has a higher nominal resolution of 3.2 Å. The processing strategy is described in Supplementary Fig. 1.

### Model building and refinement

The quality of the micelle-subtracted density map was sufficient to build a model manually in COOT[63,64]. Model building for the S3E-LysoPS bound GPR34-$G_i$ complex was facilitated by the predicted GPR34 model in AlphaFold Protein Structure Database (https://alphafold.ebi.ac.uk/entry/Q9UPC5) and the cryo-EM structure of the μOR–$G_i$ complex (PDB 6DDE). We manually modelled GPR34, the $G_i$ heterotrimer, and scFv16 into the map by jiggle fit using COOT. The TM6 helix was manually fit into the density in COOT. We then manually readjusted the model into the density map using COOT and refined it using phenix.real_space_refine (v.1.19)[65,66], with secondary-structure restraints imposed using phenix.secondary_structure_restraints. Finally, we refined the model using servalcat[67]. Model building for the M1-bound GPR34-$G_i$ complex was initiated from the S3E-bound structure and followed the same procedure.

### Molecular dynamics (MD) simulations and docking simulation

The coordinate of M1- and S3E-LysoPS-bound GPR34 pdb structures were imported into the Maestro2019-3 and processed using Protein Preparation Wizard (Schrödinger, LLC, New York, NY, USA). The N- and C-termini were omitted and capped with N-acetyl and N-methyl amide groups, respectively. Protonation states were optimized using PROPKA, and the whole structure of the ligand–receptor complex was minimized locally (force field, OPLS3e[68,69]) before the solvent model was added. Initial ligand–receptor complex models were embedded in a 1-palmitoyl-2-oleoyl-*sn*-glycero-3-phosphatidylcholine (POPC) membrane. The system was solvated with TIP3P water molecules and neutralized by adding 0.15 M NaCl. The prepared system contains ~40,000

atoms in total. The system was first subjected to preparation MD, followed by 1 μs of production MD simulation in the GPU Desmond suite (v.3.8.5.19)[70]. The relaxation protocol contains 6 stages: (1) Simulate in the NVT ensemble using Brownian dynamics for 50 ps under a temperature of 10 K, with restraints on the solute heavy atoms (50 kcal mol⁻¹ Å⁻²). (2) Simulate in the NVT ensemble using Brownian dynamics for 20 ps under 100 K, with a pressure of 1000 bar and restraints on the solute and membrane heavy atoms with a force constant of 50 kcal mol⁻¹ Å⁻². (3) Simulate in the NPγT ensemble using the MTK (Martyna-Tobias-Klein) method for 100 ps under 100 K, with a pressure of 1000 bar, restraints on the solute heavy atoms with a force constant of 10 kcal mol⁻¹ Å⁻², and restraints on the membrane N and P atoms in the z direction with a force constant of 2 kcal mol⁻¹ Å⁻². (4) Simulate in the NPγT ensemble using the MTK method for 150 ps, with the temperature gradually increasing from 100 K to 300 K, at a pressure of 100 bar. The restraints are gradually reduced to 0. From stages 2 to 4, a Gaussian biasing force is applied so the waters do not permeate the membrane. (5) Simulate in the NVT ensemble using the NH (Nosé-Hoover) method for 50 ps under 300 K, with restraints on the protein backbone and the ligand heavy atoms with a force constant of 5 kcal mol⁻¹ Å⁻². (6) Simulate in the NVT ensemble using the NH method for 50 ps under 300 K without any restraints. The production MD simulations were performed in the NPγT ensemble at 300 K using Langevin dynamics, and long-range electrostatic interactions were computed using the u-series algorithm[71].

For *sn*-1 and *sn*-2 LysoPS-bound GPR34, the proper 3D conformation and ionization states of ligands (*sn*-1 and *sn*-2 LysoPS) generated using LigPrep, under the OPLS3e force field[68,69], were used for ligand docking. The molecules were docked to the grid generated from the S3E-LysoPS-bound cryo-EM structure, using the Glide SP mode, and strain correction was applied in the post-docking score. As a result, the docking pose with the best glide score was selected for each ligand and subjected to MD simulation, following the same protocol used for the M1- and S3E-LysoPS-bound structures, outlined above. The average structure of each complex was calculated from the average coordinates of ligand–protein complex atoms during 1-μs MD simulations.

### Reporting summary

Further information on research design is available in the Nature Portfolio Reporting Summary linked to this article.

## Data availability

Density maps and structure coordinates have been deposited in the Electron Microscopy Data Bank (EMDB) and the PDB, with accession codes EMD-38215 and PDB 8XBE for the S3E-LysoPS-GPR34-$G_i$ complex; EMD-38217 and PDB 8XBG for the S3E-LysoPS-GPR34-$G_i$ complex (Receptor focused); EMD-38218 and PDB 8XBH for the M1-GPR34-$G_i$ complex; EMD-38219 and PDB 8XBI for the M1-GPR34-$G_i$ complex (Receptor focused). The dynamics data and the simulation protocols were uploaded in GPCRmd[72] (https://www.gpcrmd.org/), with dynamic IDs 1741 for the S3E-LysoPS-GPR34 complex; 1742 for the M1-GPR34 complex; 1743 for the *sn*1-18:1 LysoPS-GPR34 complex; 1744 for the *sn*2-18:1 LysoPS-GPR34 complex, and 1745 for the S3E-LysoPS_noamine-GPR34 complex. Source data are provided as a Source Data file. Source data are provided with this paper.

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

## Acknowledgements

We thank K. Ogomori and C. Harada for technical assistance. This work was supported by grants from the JSPS KAKENHI, grant numbers 21H05037 (O.N.), 22K19371 and 22H02751 (W.S.), and 22H00438 (J.A.); the ONO Medical Research Foundation (W.S.); the Kao Foundation for Arts and Sciences (W.S.); the Takeda Science Foundation (W.S.); the Uehara Memorial Foundation (W.S.); the Lotte Foundation (W,S,), the Kobayashi Foundation, Osaka, Japan (T.O.); the KOSÉ Cosmetology Research Foundation (T.O.); the Japan Agency for Medical Research and Development (AMED), grant numbers JP233fa627001 (O.N.); 22ck0106533h0003 (J.A., O.N. and T.O.); 21gm0010004h9905 (J.A.); and the Platform Project for Supporting Drug Discovery and Life Science Research (Basis for Supporting Innovative Drug Discovery and Life Science Research (BINDS)) from AMED, grant numbers JP22ama121002 (support number 3272, O.N.) and JP22ama121012 (supporting number 5529 and 5530, J.A.). L.C. wishes to also thank the Otsuka Toshimi Scholarship Foundation, Osaka, Japan.

## Author contributions

T.I. expressed, purified, and prepared grids of the S3E-LysoPS bound GPR34-Gi complex, with assistance from K.K. and H.O. Y.K. performed the cryo-EM analysis of the S3E-LysoPS-bound complex. T.I. and W.S. performed single-particle analysis and model building of the S3E-LysoPS-bound complex. R. K. performed the structural study of the M1-bound receptor, with assistance from T.T., F. K. S and W.S. W.S. initially screened and established the protocol for sample preparation. A.U., S.Y. F.H. and J.O. performed the TGFα shedding assay, the cAMP assay, and LC-MS/MS analysis, and wrote part of the manuscript. H.K. assisted with lipid preparation for the assay. J.A. supervised most of the biological experiments and wrote the biological part of the manuscript. L.C. and T.O. performed and oversaw the MD simulations and wrote the simulation part of the manuscript. The manuscript was mainly prepared by W.S., R.K., T.I., T.O. and J. A., with input from all authors and assistance from O.N.

## Competing interests

O.N. is a co-founder and scientific advisor for Curreio. All other authors declare no competing interests.
