## [Peer Review File · Nature Communications]

Structural basis for lysophosphatidylserine recognition by GPR34REVIEWER COMMENTS

Reviewer #1 (Remarks to the Author):

Izume et al. present the cryo-electron microscopy structures of the human GPR34-Gi complex recently identified to have an immunomodulatory role. The authors find that the ligand-binding pocket of GPR34 is laterally open towards the membrane, allowing for lipidic agonists to enter the cavity laterally. Two ligands, S3E-LysoPS and M1, were experimentally solved in complex with the GPR34. Both structures showed that a charged residue cluster in the orthosteric site recognizes the amine and carboxylate groups of the serine moiety. Molecular dynamics simulations further explained the LysoPS-regioselectivity of GPR34. Ultimately, their work suggests that the chemically unstable 2-acyl LysoPS is the physiological ligand for GPR34, indicating its short signal duration.

All in all, this work provides novel insights into lipid binding events for GPCRs. However, several aspects of this work require clarifications and improvements:

Major comments:

1. Authors write “Within the hydrophilic pocket, alanine mutants of the four aromatic residues Y1353.33, F205ECL2, Y207ECL2, and Y2896.58, which are critical for hydrophilic pocket formation, reduced potency of S3E-LysoPS (pEC50) by over 100-fold”

I find it confusing that the hydrophobic phenylalanine (F205) contributes to a hydrophilic pocket and upon mutation to alanine (F205A) dramatically reduces potency (Fig. 3g). Authors lack the description of 3g in the figure 3 legend (the g panel is also absent in the figure 4 legend). What does the abbreviation “UD” stand for?

2. Mutations. In Figures 3G and 4G, the reader can appreciate a dramatic change of potency upon a K128A mutation comparing the S3E-LysoPS and M1 complex. However, simulations suggest that K128 has similar contact frequencies for S3E-LysoP (33%) and M1 (31%).

The authors should discuss this observation.

2. Ligand access. Authors conclude that the ligand enters the receptor from the lateral membrane channel rather than through the extracellular opening based on indirect assays (e.g., ligand displacement from the membrane). A more direct approach would be to mutate small channel-flanking residues to bulky residues (e.g., G185F/W and A182F/W). This would completely/partially close the channel with consequences for receptor activation.

3. Authors write: "Further, a PS-PLA1 S166A mutant only weakly activates GPR34 (Extended Data Fig. 6a)." The rationale why authors use a PS-PLA1 S166A mutant is missing in the main text. Please also correct "1" to subscript in PS-PLA1.

4. Authors write that "Results further suggest that anionic charge repulsion between the phosphate (PO₄⁻) and serine CO₂⁻ moieties could contribute to a preference for the U-shaped conformation of the hydrophilic portions of these ligands." It is not clear from which data this was concluded (Figure, Table?). Please try to provide more evidence here.

5. Authors observe that

(i) "MD simulations also indicate that M1 forms significant stable interactions with the receptor (Extended Data Fig. 7b,c) and dynamically freezes movement of the extracellular half of TM4 (RMSFA182-I191 <0.8 Å) (Extended Data Fig. 7d), whereas S3E-LysoPS allows some movement of this helix (RMSFA182-I191 = 1.0–1.8 Å) (Extended Data Fig. 7e)."

(ii) "Additionally, distances between residues in TM4 and TM5 are constant in the M1-bound receptor but varied in the S3E-bound receptor."

It is not discussed how those differences link to the superagonist activity of M1.

6. One of my major concerns is: Can authors exclude that the superagonist activity of M1 is not completely related to its metabolic stability in contrast to other tested compounds? Please comment on this.

7. Regioselectivity of LysoPS species binding to GPR34

It would be good if the authors could provide binding affinities for the tested ligands. This could help us understand if the regioselectivity is related to a loss in binding affinity or if not, to the stabilization of differential receptor states that cannot activate G_i as efficiently.

8. MD simulation. It is not clear when reading the manuscript if simulations were carried out with or without the Gi protein. I assume it was done without. However, if this is the case, authors should rationalize why they did not simulate the entire ligand-GPR34-Gi complex. On the other hand, it would be very interesting to characterize the entire complex in terms of differences in contact frequencies in the GPR34-Gi protein interface.

8. Data sharing for reproducibility and transparency.

The dynamics data and the simulation protocols should be made available to the community for reproducibility and data transparency. The specific repository GPCRmd (www.gpcrmd.org) has been created for this purpose.

Reviewer #2 (Remarks to the Author):

The manuscript presents two cryo-EM structures of GPR34-Gi protein complexes with bound ligands S3E-LysoPS and M1, which are analogs of sn-3 18:1 LysoPS. The authors demonstrate the functional roles of receptor residues involved in S3E-LysoPS and M1 bindings through mutation studies. Furthermore, they observe that BSA can decrease the activation of GPR34 by LysoPS or M1. MD simulations support the ligand poses in the complex structures and demonstrate the ligand specificity. The authors describe the interactions between GPR34 and Gi protein and provide insights into ligand specificity. Overall, the authors provide useful structural information related to GPR34 and LysoPS analogs.

Main comments:

1. The validation reports indicate that >90% of the modeled residues lie outside their corresponding density maps, and the deposited models do not match their maps. Therefore, the authors should consider re-depositing their final data to PDB and re-running the validation reports.
2. The resolutions of the structures do not allow unambiguous modeling of the two ligands, and some of the ligand features might be fit to noise signals. Detailed interactions illustrated in Figure 3 and Figure 4 rely on a more accurate model built upon better density maps, and increasing particle numbers may be helpful in this regard.

Specific comments:

1. The authors should clarify that their data show both lateral and extracellular access to the ligand-binding pocket, rather than emphasizing only lateral access. Furthermore, the data in Fig. S6 does not provide conclusive proof of lateral access, but is consistent with the hypothesis.

2. The statement that "the Emax value for M1 is higher than for all agonists, suggesting it acts as a superagonist" is not entirely accurate, as the curves for sn-2 LysoPS (18:1) did not reach saturation. Therefore, it is uncertain whether M1 is a superagonist.
3. The authors should provide more evidence to support the claim that hydrophobic interactions make M1 a superagonist.
4. It is unclear whether the authors meant to refer to LPA1-6 receptors in line 47.
5. It lacks statistical analysis in Fig. S6F.
6. Line 307: " where TM6 and TM7 move farther from the center of the TM bundle". In class A GPCRs, the activation moves TM6 away from the center of the TM bundle, but moves TM7 closer to the center of the TM bundle. The authors should clarify this.

Reviewer #3 (Remarks to the Author):

Izume et al. report the cryo-EM structure for GPR34, a receptor for lysophosphatidylserine, which was initially defined by one of the authors (Aoki). While reasonably executed, there are a number of issues that should be addressed – preferably by experimentation - to improve the manuscript, which are detailed numerically below. Of particular import is the comparison of the current structures to the admittedly very recent cryo-EM structure on GPR174 (Liang et al., 2023): <https://www.nature.com/articles/s41467-023-36575-0>. There is no doubt that Liang et al. was known to the authors, and it must now be carefully compared to the current submission that reassuringly, shares similarities as well as involves different treatments, making Izume et al., a valuable contribution for the field, once modifications are made to the current version.

1. Figure 1A and related text: "sn-3 LysoPS" is confusing, because it is generally equal to "sn-1 LysoPS" based on the definition of stereospecific numbering. The authors should change the naming to something different and provide IUPAC naming for all the compounds to avoid confusion.
2. Figure 2: Neither S3E-LysoPS nor M1 are natural ligands, but an agonist and a superagonist, respectively. Therefore, the presentation should be re-organized to compare the agonist vs. the superagonist.
3. Page 6 line 110: Disulfide bond C46N-ter-C2997.25 is not a typical, since only ~15% class A GPCRs (~50) have C7.25: please modify accordingly.
4. Figure 3G: Mutagenesis lacks a comparison between synthetic ligands vs. endogenous ligands (1-acyl and 2-acyl LysoPS) to determine the binding mode of natural ligands. This should be presented.
5. Figure 3G, Page 7, line 146~ and Page 8, line 161~: The authors mentioned "there is no strict spatial requirement to accommodate the acyl chain." This statement appears to indicate that acyl chain is not

necessary for activating GPR34. Indeed, 2-acyl 14:0 LysoPS seems to be a better agonist than 18:1 [PMID: 22343749]. Conversely, are LysoPS species containing very long acyl chain (>22) more potent than shorter one? Please clarify, preferably via experimentation.

6. Page 8 line 165~ and Figure S6: The data partially support, but do not fully support, lateral diffusion of LysoPS to activate GPR34.

a. PS-PLA1 treatment produces only twice as much LysoPS as non-stimulation (Fig. S6B). GPCR activation (%AP-TGF β release) induced by PS-PLA1 is ~5% (Fig. 6SA) that is equivalent to 100 nM of sn-2 LysoPS (Figure 1B). This suggests the presence of ~50 nM LysoPS in unstimulated cells, which has already induced ~5% GPCR activation. The authors need to clarify this point.

b. PS is mainly located in the inner leaflet of the plasma membrane. How does the PS-PLA1 access PS and efficiently produce LysoPS to activate GPR34 when it is applied extracellularly? The authors should provide a dose response of recombinant PS-PLA1 in shedding, cAMP and LysoPS levels.

c. PS-PLA1 sensitivity of GPR34 expressing cells may be enhanced in apoptotic cells, which can be easily addressed experimentally and provide insights into biology.

d. It is unclear what solvent was used for dissolving LysoPS and other compounds for shedding and cAMP assays in Fig 1, 3G, 4E, S5, Table S1, S2. What percentage of LysoPS (free form vs. in the presence of BSA) is incorporated into the plasma membrane and is remained extracellularly?

e. Can GPR34 responses to PS-PLA1 or LysoPS be blocked by inhibiting membrane fluidity?

7. Figure 4: Since there is no biological/clinical relevance and scientific premise with M1, it is unclear about the significance of the structure in complex with M1. This should be addressed, preferably by experimentation.

8. Figure 5: Y2896.58 is one of the key amino acid residues identified by mutagenesis that is missing in the MD simulation (Fig. 5A, B), reducing the validity of the MD simulation. The simulation needs to be re-run to consider this residue.

9. Page 14, line 313: suggest a change to the sentence to read "...a characteristic shared among..." since a "unique characteristic" cannot be shared if it is unique.

10. Page 14, line 316: The uniqueness of GPR34 appears to be R2866.55 rather than the E7.36 that is also found in S1P1.

11. Related to #6, considering the mutagenesis results, the importance of the L-shaped pocket seems to be only for M1. Natural ligands may not require the L-shaped pocket. Overall, the insight about hydrophobic pocket is not very convincing and should be re-written to better capture limitations to the data.

12. D151A mutation of GPR34 has been reported to associated with lymphoma and seems to be essential for interacting with G β i protein (Fig. 6C). The authors should discuss this in the context of the biological and clinical relevance of GPR34.

13. A further possible biological feature might be the flipping of PS to provide biological ligand substrate to GPR34 and other lysoPS receptors, best characterized during apoptosis but now seen in other settings, a link that should be discussed.

The parts of the text that have been changed are highlighted in yellow. The major changes are as follows:

- We re-analyzed the cryo-EM data using cryoSPARC and improved the resolutions of the structures. Based on the new structures, we re-performed the MD simulations. The agonist binding modes in the new structures and the result from the simulations were essentially consistent with those of the previous analyses.
- We have removed all claims and related data that M1 is a superagonist.
- To further investigate the lateral access, we analyzed three mutant GPR34 constructs, A182W, G185F, and G185W, which are designed to occlude the TM4-5 gap.
- Since the cryo-EM structure of GPR174 was reported during the revision, we modified our discussion comparing GPR34, P2Y10, and GPR174

Reviewer #1 (Remarks to the Author):

2. Mutations. In Figures 3G and 4G, the reader can appreciate a dramatic change of potency upon a K128A mutation comparing the S3E-LysoPS and M1 complex. However, simulations suggest that K128 has similar contact frequencies for S3E-LysoP (33%) and M1 (31%).

The authors should discuss this observation.

Thank you for pointing out the activity difference with K128A. Indeed, we performed the MD using the currently updated protein-ligand structures (pdb files), and found a difference in the ligand-K128 interaction. Even though the interaction between K128A and the ligand is conserved in the MD simulations of both the S3E-LysoPS-bound GPR34 and M1-bound GPR34 structures (Fig. 4a,b), the interaction fraction of S3E-LysoPS is lower than that of M1 (approximately 1.2 (S3E-LysoPS) vs. 1.6 (M1)). In addition, the calculated interaction energy (Force field: OPLS3e) throughout the trajectory suggested that the binding between M1 and K128 is more stable than that between S3E-LysoPS and K128 (Supplementary Fig. 7h,i). We added the description in the supplementary discussion (last section of the supplementary information).

2. Ligand access. Authors conclude that the ligand enters the receptor from the lateral membrane channel rather than through the extracellular opening based on indirect assays (e.g., ligand displacement from the membrane). A more direct approach would be to mutate small channel-flanking residues to bulky residues (e.g., G185F/W and A182F/W). This would completely/partially close the channel with consequences for receptor activation.

Thank you for the valuable comments. As suggested, we generated four mutant GPR34 constructs, A182F/W and G185F/W, and expressed them in HEK293 cells. Since the A182F mutant was not expressed at all, only the results for A182W, G185F and G185W are shown in Supplementary Fig. 6h,i. The A182W, G185F, and G185W mutants responded to S3E-LysoPS and M1 (each 100 nm) as well as or better than the wild-type GPR34. Among the three, the activation of the G185F mutant by the recombinant PS-PLA₁ protein was comparable to that of wild-type GPR34. By contrast, neither A182W nor G185W was activated by the recombinant PS-PLA₁ protein. The facts that A182W and G185W were activated by S3E-LysoPS but not by PS-PLA₁ can be interpreted as follows: The bulky side chain of W(Trp) prevented the ligand's lateral access but not its access from outside of the cell. Our preliminary findings showed that KO mice of GPR34 and PS-PLA₁ had somewhat similar phenotypes in tumor-bearing and antigen immunization models. Therefore, PS-PLA₁ may be the only LysoPS-producing enzyme that provides the ligand to GPR34. Accordingly, although LysoPS can access GPR34 from both the outside of the cell and laterally through the membrane, the LysoPS produced by PS-PLA₁ in the plasma membrane of GPR34-expressing cells migrates laterally in the plane of the plasma membrane to access GPR34. Compared to lysophospholipid mediators such as S1P and LPA, LysoPS is rarely present in the blood. This may also support the lateral ligand access model. We added the description of the mutational analysis (line 182-194).

3. Authors write: "Further, a PS-PLA1 S166A mutant only weakly activates GPR34 (Extended Data Fig. 6a)." The rationale why authors use a PS-PLA1 S166A mutant is missing in the main text. Please also correct "1" to subscript in PS-PLA1.

Thanks for the comments. First, "1" in PS-PLA1 has been corrected with a subscript. The PS-PLA₁ S166A mutant protein has no enzyme activity and thus is the ultimate negative control for wild-type PS-PLA₁. We added a statement about this in the text (line 196).

4. Authors write that "Results further suggest that anionic charge repulsion between the phosphate (PO⁻) and serine CO₂⁻ moieties could contribute to a preference for the U-shaped conformation of the hydrophilic portions of these ligands." It is not clear from which data this was concluded (Figure, Table?). Please try to provide more evidence here.

Thank you for the comment. To provide a detailed explanation of the anionic charge repulsion between the phosphate (PO⁻) and serine CO₂⁻ moieties of LysoPS derivatives, we conducted metadynamics simulations of the ligand molecules (M1 and S3E-LsoPS). We obtained the

energy landscapes of both ligands in water with respect to two dihedral angles (X-X-Y-Z and Z-Y-W-V) changing the conformation of phosphoserine (Supplementary Fig. 7c-e). As suggested by the results, the energy minimums showed U-shaped conformations (in which the two anionic centers are apart), which are consistent with the binding conformation of each ligand to GPR34 (Fig. 2e, Fig. 3b). We mentioned about this in the main text (line 227-228) and the figure legend of Supplementary Fig. 7c-e.

5. Authors observe that

(i) “MD simulations also indicate that M1 forms significant stable interactions with the receptor (Extended Data Fig. 7b,c) and dynamically freezes movement of the extracellular half of TM4 (RMSFA182-I191 <0.8 Å) (Extended Data Fig. 7d), whereas S3E-LysoPS allows some movement of this helix (RMSFA182-I191 = 1.0–1.8 Å) (Extended Data Fig. 7e).”

(ii) “Additionally, distances between residues in TM4 and TM5 are constant in the M1-bound receptor but varied in the S3E-bound receptor.”

It is not discussed how those differences link to the superagonist activity of M1.

6. One of my major concerns is: Can authors exclude that the superagonist activity of M1 is not completely related to its metabolic stability in contrast to other tested compounds? Please comment on this.

Thank you for the comments. We have removed all claims that M1 is a superagonist from the manuscript.

7. Regioselectivity of LysoPS species binding to GPR34

It would be good if the authors could provide binding affinities for the tested ligands. This could help us understand if the regioselectivity is related to a loss in binding affinity or if not, to the stabilization of differential receptor states that cannot activate Gi as efficiently.

Thank you for the critical comments. In general, binding experiments of lysophospholipid ligands to GPCR receptors are extremely difficult. This is because, compared to water-soluble ligands, lysophospholipids are highly hydrophobic, and have a strong tendency to associate and non-specifically bind to the cell membrane. In addition, high-specificity radiolabeled ligands are required for ligand binding experiments, which are unavailable for S3E-LysoPS and M1. Therefore, binding experiments could not be performed in this study.

8. MD simulation. It is not clear when reading the manuscript if simulations were carried out with or without the Gi protein. I assume it was done without. However, if this is the case, authors should rationalize why they did not simulate the entire ligand-GPR34-Gi complex. On the other hand, it would be very interesting to

characterize the entire complex in terms of differences in contact frequencies in the GPR34-Gi protein interface.

Thank you for pointing this out. The MD simulation was conducted without the Gi protein because of computation resource limitations for the large molecular weight complex. Unfortunately, as it is difficult to perform demanding calculations such as those required for the GPCR-Gi complex with our current hardware, we decided to conduct the MD simulation only for the ligand-GPCR complex.

8. Data sharing for reproducibility and transparency.

The dynamics data and the simulation protocols should be made available to the community for reproducibility and data transparency. The specific repository GPCRmd (www.gpcrmd.org) has been created for this purpose.

Thank you for the advice. After the paper is accepted and the data to be published is finalized, we will soon upload the dynamics data and the simulation protocols on this site (line 415-416).

Reviewer #2 (Remarks to the Author):

Main comments:

1. The validation reports indicate that >90% of the modeled residues lie outside their corresponding density maps, and the deposited models do not match their maps. Therefore, the authors should consider re-depositing their final data to PDB and re-running the validation reports.

We re-deposited the files and obtained new validation reports.

2. The resolutions of the structures do not allow unambiguous modeling of the two ligands, and some of the ligand features might be fit to noise signals. Detailed interactions illustrated in Figure 3 and Figure 4 rely on a more accurate model built upon better density maps, and increasing particle numbers may be helpful in this regard.

According to the suggestion, we re-analyzed the cryo-EM data using cryoSPARC. We improved the resolutions of the M1-bound structures from 3.2 Å to 2.8 Å and the S3E-bound structures from 3.4 Å to 3.2 Å (Supplementary Figs 1, 2, and Supplementary Table 1). Thus, the densities of the agonists also greatly improved (Fig. 1b, c).

Specific comments:

1. The authors should clarify that their data show both lateral and extracellular access to the ligand-binding pocket, rather than emphasizing only lateral access. Furthermore, the data in Fig. S6 does not provide conclusive proof of lateral access, but is consistent with the hypothesis.

Thank you for the comments. We agree with your interpretation of our data, and suggested that LysoPS could access the ligand-binding pocket both laterally via the membrane and extracellularly from the outside of the cell. In the revised manuscript, we performed additional experiments (Supplementary Fig. 6h,i) using some GPR34 mutants (A182W, G185F, and G185W mutants), for which only the lateral access might be suppressed. The results showed that the activation by PS-PLA₁, a natural LysoPS-producing extracellular enzyme, was only inhibited in the A182W and G185W mutants. This confirmed that GPR34 had two ligand access routes. Our preliminary findings showed that KO mice of GPR34 and PS-PLA₁ had somewhat similar phenotypes in tumor-bearing and antigen immunization models. Therefore, PS-PLA₁ may be the only LysoPS-producing enzyme that provides the ligand to GPR34. Accordingly, although LysoPS can access GPR34 from the outside of the cell and laterally through the membrane, the LysoPS produced by PS-PLA₁ in the plasma membrane of GPR34-expressing cells migrates laterally in the plane of the plasma membrane to access GPR34. Compared to lysophospholipid mediators such as S1P and LPA, LysoPS is rarely present in the blood. This may also support the lateral ligand access model. We added the description of the mutational analysis (line 182-194).

2. The statement that "the Emax value for M1 is higher than for all agonists, suggesting it acts as a superagonist" is not entirely accurate, as the curves for sn-2 LysoPS (18:1) did not reach saturation. Therefore, it is uncertain whether M1 is a superagonist.

3. The authors should provide more evidence to support the claim that hydrophobic interactions make M1 a superagonist.

Thank you for the comments. We have removed all claims that M1 is a superagonist from the manuscript.

4. It is unclear whether the authors meant to refer to LPA1-6 receptors in line 47.

We simplified the sentence to "This finding prompted Makide *et al*¹⁴. to propose renaming GPR34 as LPS₁ or LPSR1, as lysophosphatidic acid receptors (LPA₁₋₆) (line 45-46).

5. It lacks statistical analysis in Fig. S6F.

We performed the statistical analysis and revised Supplementary Fig. 6f.

6. Line 307: “ where TM6 and TM7 move farther from the center of the TM bundle”. In class A GPCRs, the activation moves TM6 away from the center of the TM bundle, but moves TM7 closer to the center of the TM bundle. The authors should clarify this.

Thank you for the comments. We have removed all claims that M1 is a superagonist from the manuscript.

Reviewer #3 (Remarks to the Author):

Izume et al. report the cryo-EM structure for GPR34, a receptor for lysophosphatidylserine, which was initially defined by one of the authors (Aoki). While reasonably executed, there are a number of issues that should be addressed – preferably by experimentation - to improve the manuscript, which are detailed numerically below. Of particular import is the comparison of the current structures to the admittedly very recent cryo-EM structure on GPR174 (Liang et al., 2023): <https://www.nature.com/articles/s41467-023-36575-0>. There is no doubt that Liang et al. was known to the authors, and it must now be carefully compared to the current submission that reassuringly, shares similarities as well as involves different treatments, making Izume et al., a valuable contribution for the field, once modifications are made to the current version.

According to the suggestion, we revised the discussion of the structural comparison of the GPR34, P2Y10, and GPR174 (line 358-378).

1. Figure 1A and related text: “sn-3 LysoPS” is confusing, because it is generally equal to “sn-1 LysoPS” based on the definition of stereospecific numbering. The authors should change the naming to something different and provide IUPAC naming for all the compounds to avoid confusion.

Thank you for the comment. We added the IUPAC naming with the definitions in the figure legend of Fig. 1a.

2. Figure 2: Neither S3E-LysoPS nor M1 are natural ligands, but an agonist and a superagonist, respectively. Therefore, the presentation should be re-organized to compare the agonist vs. the superagonist.

Thank you for the comments. We have removed all claims that M1 is a superagonist from the manuscript.

3. Page 6 line 110: Disulfide bond C46N-ter-C2997.25 is not a typical, since only ~15% class A GPCRs (~50) have C7.25: please modify accordingly.

According to the suggestion, we corrected the sentence to “which is conserved in 15% of class A GPCRs” (line 106-107).

4. Figure 3G: Mutagenesis lacks a comparison between synthetic ligands vs. endogenous ligands (1-acyl and 2-acyl LysoPS) to determine the binding mode of natural ligands. This should be presented.

In general, *sn*-2 lysophospholipids quickly convert to *sn*-1 lysophospholipids. This conversion is known to be inhibited by weak acidity, but theoretically, 100% pure *sn*-2 and *sn*-1 lysophospholipids cannot be obtained. Even when *sn*-2 lysophospholipids are produced from phospholipids by the PS-PLA₁ reaction and the solution is quickly acidified, the purity of *sn*-2 lysophospholipids is only around 90%, and *sn*-1 lysophospholipids are present at ~10%. Therefore, the proposed experiment using *sn*-2 and *sn*-1 lysophospholipids cannot be carried out. We added the description (line 250-258) and the LC-MS analysis of the lysoPS product by PS-PLA₁ (Supplementary Fig. 8e-g).

5. Figure 3G, Page 7, line 146~ and Page 8, line 161~: The authors mentioned “there is no strict spatial requirement to accommodate the acyl chain.” This statement appears to indicate that acyl chain is not necessary for activating GPR34. Indeed, 2-acyl 14:0 LysoPS seems to be a better agonist than 18:1 [PMID: 22343749]. Conversely, are LysoPS species containing very long acyl chain (>22) more potent than shorter one? Please clarify, preferably via experimentation.

As you noted, we previously reported the acyl chain-dependency of GPR34 activation (PMID: 22343749). The results revealed that LysoPS with oleic acid at the *sn*-2 position was the best agonist. In addition, we have data showing that LysoPS with polyunsaturated fatty acids, including DHA, at the *sn*-2 position was by far the best. We are planning to use the ligand specificity data in another paper and will not show the data here.

6. Page 8 line 165~ and Figure S6: The data partially support, but do not fully support, lateral diffusion of LysoPS to activate GPR34.

a. PS-PLA₁ treatment produces only twice as much LysoPS as non-stimulation (Fig. S6B). GPCR activation (%AP-TGF α release) induced by PS-PLA₁ is ~5% (Fig. 6SA) that is equivalent to 100 nM of *sn*-2 LysoPS (Figure 1B). This suggests the presence of ~50 nM LysoPS in unstimulated cells, which has already induced ~5% GPCR activation. The authors need to clarify this point.

Thank you for the critical comments. Most LysoPS is found intracellularly, and this intracellular LysoPS does not activate the receptor (GPR34). However, the LysoPS produced by PS-PLA₁ can be extracted by the addition of BSA and is therefore considered to be present

on the cell surface. Thus, the 2-fold increase in LysoPS by the addition of a PS-PLA₁ recombinant protein on the outside of the cell is mostly at the cell surface. The addition of BSA, which pulls the cell surface LysoPS from the plasma membrane, strongly inhibits the activation of GPR34 by PS-PLA₁. Note that the LysoPS added from outside the cell as a solution is distributed in 3D in the medium, whereas the LysoPS generated by the action of PS-PLA₁ on the cell membrane surface is distributed in 2D (in the membrane). It is thus difficult to simply compare the LysoPS concentrations in solution (3D) and on the cell membrane (2D) and discuss their activities. From our analyses, the concentrations of LysoPS in plasma and the supernatants of various cultured cells are very low; *i.e.*, sub nM, and are too low to activate the receptor. Therefore, we believe that LysoPS concentrations of several tens of nM are rarely present in the extracellular medium and that the LysoPS is produced at the plasma membrane and supplied to the receptor by lateral access in the membrane, where its concentration is expected to be much higher.

b. PS is mainly located in the inner leaflet of the plasma membrane. How does the PS-PLA1 access PS and efficiently produce LysoPS to activate GPR34 when it is applied extracellularly? The authors should provide a dose response of recombinant PS-PLA1 in shedding, cAMP and LysoPS levels.

Thank you for the important point. Most PS is present on the cytosolic surface of the plasma membrane. However, in almost all cells, PS is exposed on the cell surface to some extent, and this PS exposed on the outside of the cell is quickly flipped into the cell. Certain types of cells; *e.g.*, apoptotic cells, cells that have undergone cell death, cancer cells, and activated immune system cells (neutrophils, T, B, DC, macrophages, etc.) have large amounts of PS exposed on their cell surfaces. In this context, we have previously shown that PS-PLA₁ hydrolyzed PS on the plasma membrane of neutrophils to produce LysoPS (Phosphatidylserine-specific phospholipase A1 stimulates histamine release from rat. Hosono H, Aoki J, Nagai Y, Bandoh K, Ishida M, Taguchi R, Arai H, Inoue K. Biol Chem. 2001 Aug 10;276(32):29664-70. doi: 10.1074/jbc.M104597200). We already refer to the paper as reference # 35.

c. PS-PLA1 sensitivity of GPR34 expressing cells may be enhanced in apoptotic cells, which can be easily addressed experimentally and provide insights into biology.

T cells and neutrophils leached into inflammatory sites have large amounts of PS exposed on their cell surfaces. We previously reported that the addition of recombinant PS-PLA₁ to these

cells greatly enhanced the production of LysoPS, and thus activated the mast cell LysoPS receptor (Hosono et al., above). In this paper, we also refer to Hosono et al.

d. It is unclear what solvent was used for dissolving LysoPS and other compounds for shedding and cAMP assays in Fig 1, 3G, 4E, S5, Table S1, S2. What percentage of LysoPS (free form vs. in the presence of BSA) is incorporated into the plasma membrane and is remained extracellularly?

Thank you for the comment. The revised manuscript describes how the LysoPS was dissolved in detail in the Materials & methods (line 460-480).

e. Can GPR34 responses to PS-PLA1 or LysoPS be blocked by inhibiting membrane fluidity?

In general, plasma membrane fluidity depends on the degree of unsaturation of plasma membrane phospholipids. In our preliminary experiments, the signals from many GPCRs, including GPR34, did not change much when unsaturated fatty acids were added to the cells to increase the fluidity of the plasma membrane. As we believe that this type of experiment is far from the essence of this paper, we will not include its results in the text.

7. Figure 4: Since there is no biological/clinical relevance and scientific premise with M1, it is unclear about the significance of the structure in complex with M1. This should be addressed, preferably by experimentation.

In our preliminary unpublished results, the administration of a LysoPS analog (M1) was shown to change the macrophages in cancer tissue from the M2- to M1-type in a mouse carcinoma model, resulting in a decrease in cancer tissue weight. On the other hand, this paper discusses the ligand recognition mechanism of GPR34. Moreover, we need a thorough examination of the biological activity of the LysoPS analog. We plan to publish it in our next paper. Thus, the data showing the effects of a LysoPS analog on increasing M1-type macrophages and shrinking cancer tissues are not shown in this paper.

8. Figure 5: Y2896.58 is one of the key amino acid residues identified by mutagenesis that is missing in the MD simulation (Fig. 5A, B), reducing the validity of the MD simulation. The simulation needs to be re-run to consider this residue.

Thank you for the suggestion. We reran the MD simulation with the current updated structure. In the new MD simulation, we found that Y289^{6.58} formed hydrophilic interactions with the serine moieties of ligands in the M1- and S3E-LysoPS-bound GPR34 structures. These interactions are stable, but occur in a low-frequency manner (Fig. 4a,b). In addition to direct

contact with the ligand, Y289^{6.58} also reinforces the binding pocket by tilted T-shaped pi-pi stacking with Y207^{ECL2} (Supplementary Fig. 7j,k). Thus, the Y289A mutant should show dramatically reduced activity in the mutagenesis study. We added the description of the interaction between Y207^{ECL2} and Y289^{6.58} (lines 125) and an explanation of the MD simulation in the supplementary discussion (last section of the supplementary information).

9. Page 14, line 313: suggest a change to the sentence to read "...a characteristic shared among..." since a "unique characteristic" cannot be shared if it is unique.

According to the suggestion, we revised the sentence as "a characteristic shared among some types of lipid-sensing GPCRs such as LPA₆" (line 323-324).

10. Page 14, line 316: The uniqueness of GPR34 appears to be R2866.55 rather than the E7.36 that is also found in S1P1.

Thank you for the comment. However, E^{7.36} is also found in S1P1. We wanted to emphasize here that both the carboxyl and amine groups of serine are recognized by the charging amino acids of the receptor and that we consider this characteristic to be part of the uniqueness of GPR34.

11. Related to #6, considering the mutagenesis results, the importance of the L-shaped pocket seems to be only for M1. Natural ligands may not require the L-shaped pocket. Overall, the insight about hydrophobic pocket is not very convincing and should be re-written to better capture limitations to the data.

Thank you for the comment. We share the same understanding, but again, we emphasized that throughout the manuscript.

12. D151A mutation of GPR34 has been reported to associated with lymphoma and seems to be essential for interacting with Gai protein (Fig. 6C). The authors should discuss this in the context of the biological and clinical relevance of GPR34.

Thank you for the suggestion. We added a discussion about the D151A mutation in the supplementary discussion (last section of the supplementary information).

13. A further possible biological feature might be the flipping of PS to provide biological ligand substrate to GPR34 and other lysoPS receptors, best characterized during apoptosis but now seen in other settings, a link that should be discussed.

Thank you for the important point. Most PS is present on the cytosolic surface of the plasma membrane. However, in almost all cells, PS is exposed to the cell surface to some extent, and this PS exposed on the outside of the cell is quickly flipped back into the cytoplasmic side. Certain types of cells; *e.g.*, apoptotic cells, cells that have undergone cell death, cancer cells, and activated immune system cells (neutrophils, T, B, DC, macrophages, etc.) have large amounts of PS exposed on their cell surfaces. In this context, we previously reported that PS-PLA₁ hydrolyzed the PS on the plasma membrane of neutrophils to produce LysoPS (Phosphatidylserine-specific phospholipase A1 stimulates histamine release from rat. Hosono H, Aoki J, Nagai Y, Bando K, Ishida M, Taguchi R, Arai H, Inoue K. Biol Chem. 2001 Aug 10;276(32):29664-70. doi: 10.1074/jbc.M104597200). We already referred to the paper as reference #35.

REVIEWER COMMENTS

Reviewer #1 (Remarks to the Author):

After carefully considering the revisions made by the authors, I am delighted to state that all my concerns have been adequately addressed. The authors' efforts in addressing each point have resulted in a substantially improved manuscript which makes a significant contribution to the field.

Reviewer #2 (Remarks to the Author):

The authors have satisfactorily addressed most of my concerns.

Reviewer #3 (Remarks to the Author):

Although the GPR34 structure is novel, the biological significance was still not well addressed in this revised paper. The authors provided a partial response that while somewhat responsive, still lacked essential experimental data, especially for critique points #4 and #5. A summary of the critique #s and their responses follows next along with specific examples where more information is needed or highly desirable.

#0-3 addressed

#4 (Essential) not addressed,

need experimental dataset to determine biological activity of natural ligands

#5 (Essential) not addressed,

need experimental dataset to determine biological activity of natural ligands

#6a addressed

#6b (Optional) partially addressed, but not experimentally addressed.

#6c (Important) not addressed,

Experimental data will strengthen the paper in view of biology

#6d (Priority) partially addressed,

need to revise the method sections

#6e addressed

#7`12 addressed

#13 (Important) addressed, but need some experimental validation raised in #6c

Critique point #4. Figure 3G: Mutagenesis lacks a comparison between synthetic ligands vs. endogenous ligands (1-acyl and 2-acyl LysoPS) to determine the binding mode of natural ligands. This should be presented.

Authors' Response: In general, sn-2 lysophospholipids quickly convert to sn-1 lysophospholipids. This conversion is known to be inhibited by weak acidity, but theoretically, 100% pure sn-2 and sn-1 lysophospholipids cannot be obtained. Even when sn-2 lysophospholipids are produced from phospholipids by the PS-PLA1 reaction and the solution is quickly acidified, the purity of sn-2 lysophospholipids is only around 90%, and sn-1 lysophospholipids are present at ~10%. Therefore, the proposed experiment using sn-2 and sn-1 lysophospholipids cannot be carried out. We added the description (line 250-258) and the LC-MS analysis of the lysoPS product by PS-PLA1 (Supplementary Fig. 8e-g).

This response ignored the question about whether the binding mode of natural ligands are equivalent to or different from the synthetic ligands. The authors should provide experimental data like Fig 2g and 3e using natural ligands (either sn-1 or sn-2 lysoPS). Although the authors replied to #7 that "we need a thorough investigation of the biological activity of the LysoPS analog", an experimental data set comparing the binding mode of synthetic vs. natural ligands is really necessary for this manuscript.

Critique point #5. Figure 3G, Page 7, line 146~ and Page 8, line 161~: The authors mentioned "there is no strict spatial requirement to accommodate the acyl chain." This statement appears to indicate that acyl chain is not necessary for activating GPR34. Indeed, 2-acyl 14:0 LysoPS seems to be a better agonist than 18:1 [PMID:22343749]. Conversely, are LysoPS species containing very long acyl chain (>22) more potent than shorter one? Please clarify, preferably via experimentation.

Authors' Response: As you noted, we previously reported the acyl chain-dependency of GPR34 activation (PMID: 22343749). The results revealed that LysoPS with oleic acid at the sn-2 position was the best agonist. In addition, we have data showing that LysoPS with polyunsaturated fatty acids, including DHA, at the sn-2 position was by far the best. We are planning to use the ligand specificity data in another paper and will not show the data here.

Same as point #4: the ligand specificity data are important to understand the structural features in this manuscript. The paper needs to show the structures with a native ligand, not just a synthetic ligand, for biological significance.

Critique point 6c. PS-PLA1 sensitivity of GPR34 expressing cells may be enhanced in apoptotic cells, which can be easily addressed experimentally and provide insights into biology.

Authors' Response: T cells and neutrophils leached into inflammatory sites have large amounts of PS exposed on their cell surfaces. We previously reported that the addition of recombinant PS-PLA1 to these cells greatly enhanced the production of LysoPS, and thus activated the mast cell LysoPS receptor (Hosono et al., above). In this paper, we also refer to Hosono et al.

The authors proposed the lateral diffusion of lysoPS to activate GPR34 throughout the paper, yet this response supports transcellular activation but not the lateral diffusion. The current paper lacks direct evidence for lateral diffusion mechanisms. The authors should also provide experimental evidence that GPR34-expressing cells are highly activated in ongoing apoptosis states as compared to naïve conditions.

Critique point 6d. It is unclear what solvent was used for dissolving LysoPS and other compounds for shedding and cAMP assays in Fig 1, 3G, 4E, S5, Table S1, S2. What percentage of LysoPS (free form vs. in the presence of BSA) is incorporated into the plasma membrane and is remained extracellularly?

Authors' Response: Thank you for the comment. The revised manuscript describes how the LysoPS was dissolved in detail in the Materials & methods (line 460-480).

Still missing is the information on what solvent and vehicle controls were used in the cAMP, TGF- α shedding, and other assays, especially lines 491-493 and 520-523. Were LysoPS and synthetic ligands solubilized in the same vehicle? No DMSO? With or without BSA? The methods section needs to be extensively rewritten to provide sufficient information for other researchers to reproduce the experiments.

The parts of the text that have been changed are highlighted in yellow.

We only addressed the comments in referee #3 as noted by the editor.

Referee #3 ;

Critique point #4. Figure 3G: Mutagenesis lacks a comparison between synthetic ligands vs. endogenous ligands (1-acyl and 2-acyl LysoPS) to determine the binding mode of natural ligands. This should be presented.

This response ignored the question about whether the binding mode of natural ligands are equivalent to or different from the synthetic ligands. The authors should provide experimental data like Fig 2g and 3e using natural ligands (either sn-1 or sn-2 lysoPS). Although the authors replied to #7 that "we need a thorough investigation of the biological activity of the LysoPS analog", an experimental data set comparing the binding mode of synthetic vs. natural ligands is really necessary for this manuscript.

According to the Referee's comments, we have included new experimental data of mutagenesis analysis of 18 residues using natural ligands (either sn-1 or sn-2 lysoPS) (Supplementary Fig. 9), as Fig 2g and 3e towards M1 ligand. We have also added the description about the result saying "while the mutagenesis analysis of the entire ligand-binding pocket suggests that the hydrophilic pocket plays an essential role in the recognition of the polar head groups of both sn-1 and sn-2 type, as the synthetic ligands (Supplementary Fig. 9)" (lines 258 to 261).

Still missing is the information on what solvent and vehicle controls were used in the cAMP, TGF- α shedding, and other assays, especially lines 491-493 and 520-523. Were LysoPS and synthetic ligands solubilized in the same vehicle? No DMSO? With or without BSA? The methods section needs to be extensively rewritten to provide sufficient information for other researchers to reproduce the experiments.

According to the suggestion, we added the information about the solution about 0.01% BSA/HBSS (lines 495, 496, 519, and 524).